# Architect Thyself: Neural Darwinism and Self-Evolving Multimodal Networks

## Abstract

Modern deep learning architectures, particularly Vision-Language Models (VLMs), have achieved remarkable success across a wide range of multimodal tasks. However, these models are often constrained by manually engineered, static topologies with predefined architectural blueprints that limit their adaptability, diversity, and evolutionary potential. Such rigidity hampers their ability to generalize across domains, scale efficiently, and innovate beyond human design. To address these limitations, we present AI Architect Thyself, a meta-learned evolutionary framework that enables neural networks to design, diversify, and evolve their own architectures. Unlike conventional neural architecture search or fixed multimodal blueprints, our approach treats topology as a dynamic, learnable variable optimized jointly with network parameters. Our Thyself Architect introduces three key innovations: (i) Parametric Purality (PP) where multiple instantiations of diverse archetypes (e.g., Transformers, LSTMs, ResNets, Squeeze-and-Excite modules) coexist with distinct hyperparameters; (ii) a Graph Attention Router (GAR) that performs per-sample expert routing across a dynamically evolving module zoo; and (iii) a co-evolutionary hybridization engine that recombines architectural traits of high-performing ancestors to generate novel configurations beyond human design. Across 12 multimodal and vision-language benchmarks, including Hateful Memes, VQA v2.0, COCO Captions, Food-101, and Open-Images, our framework consistently surpasses state-of-the-art baselines with improvements of +0.9% to +4.1% in accuracy, AUC, and F1-Score. These results demonstrate a paradigm shift: models can evolve from engineered artifacts into self-directed, evolving organisms, advancing the frontier of autonomous machine intelligence.

## 1 Introduction

The design of neural architectures has traditionally relied on manual, trial-and-error exploration, requiring significant expertise and computational effort. Practitioners iteratively tune hyperparameters and evaluate static blueprints, a rigid process constrained by human intuition and resistant to adaptability. Neural Architecture Search (NAS) emerged to automate this pipeline; however, it too remains bounded by the need for predefined search spaces and static optimization strategies. Approaches such as reinforcement learning, evolutionary algorithms, and gradient-based methods ultimately treat architecture as a fixed hyperparameter rather than a dynamic, learnable variable.

Despite notable progress, current NAS approaches still face critical limitations. They rely on constrained, human-engineered search spaces, which restrict the discovery of novel architectures (Ouertatani et al., 2025; Lopes & Alexandre, 2025), and employ computationally expensive evaluation strategies that require full training of candidate networks (Barradas-Palmeros et al., 2025; Xun et al., 2023). In addition, most search strategies are static, lacking mechanisms to adapt or leverage prior learning (Wang & Zhu, 2024; Yang et al., 2021). Finally, existing methods fail to capture parametric diversity, neglecting the potential of multiple instantiations of architectural components with distinct hyperparameters (Ouertatani et al., 2025; Lim & Kim, 2022). These challenges naturally raise a fundamental question that we address in this paper.

*"Can a neural network learn to become its own architect, continuously evolving its internal structure to better master a task?"*

To address this question, we introduce a fully autonomous neural framework that empowers networks to self-architect, self-optimize, and continuously self-evolve. Unlike conventional NAS approaches limited by static topologies, our system engages in a co-evolutionary process guided by a meta-cognitive controller that learns not only the network parameters but also the underlying architectural principles. The core intuition is that by enabling a network to modify its own structure during training, it can discover novel, high-performing designs beyond human foresight. The controller actively monitors structural modifications that have the potential to enhance performance, internalizing effective design strategies from experience and enabling continuous refinement over time. To further enhance specialization, the system maintains a diverse ensemble of neural modules, incorporating repeated components such as Transformers along with unique internal configurations (e.g., varying attention heads, depths, or connection patterns). This modular diversity allows individual components to master distinct subproblems while collectively advancing the overall architecture's capabilities.

Extending NAS to meet the above requirements introduces several fundamental challenges, which we address through the novel methods.

(i) **Static and Inefficient Inference:** Conventional neural networks operate with a fixed structure and computational path for every input, regardless of its complexity. To overcome this limitation, we introduce a Graph Attention Router (GAR), which dynamically selects data-dependent pathways through the network. By leveraging learned attention, it activates only the most relevant expert modules for each input, enabling context-aware and computationally efficient inference.

(ii) **Limited Architectural Search Spaces:** Standard NAS methods are constrained by predefined, human-engineered search spaces, which restrict the discovery of truly novel architectures. Our Co-Evolution Engine overcomes this by employing biologically inspired modular recombination, that intelligently combines the high-performing features from existing modules to generate entirely new and diverse architectural configurations.

(iii) **Difficulty in Generating Novel Yet Effective Architectures:** Random mutations or naive search strategies often produce suboptimal or inefficient designs. We use an intelligent hybridization, a co-evolution engine that identifies successful structural motifs and strategically cross-breeds them. This guided evolutionary process accumulates "architectural wisdom," enabling the creation of innovative, high-performing designs that go beyond the limits of human-constrained search spaces.

Building on the challenges outlined above and the novel methods we use to address them, we now summarize the major contributions of our work.

- **A Framework for Autonomous Architectural Evolution:** Rather than relying on a static, manually defined architecture, we introduce a co-evolutionary hybridization engine that enables the network to design itself. This process is guided by a self-growth strategist that learns effective evolutionary policies from a replay memory of successful past modifications. By intelligently recombining the structural traits and hyperparameters of high-performing "ancestor" networks, the system generates entirely new and more effective modules. In this way, the network's topology is no longer a fixed blueprint but a dynamic variable optimized jointly with the model's weights.

- **Parametric plurality with dynamic expert routing:** We introduced the novel concept of parametric plurality, where the network builds and maintains a diverse "zoo" of specialized modules. Under this principle, even modules of the same type (e.g., multiple transformers or ResNets) are instantiated with unique hyperparameters, allowing each one to become an expert at a specific sub-task. To leverage this diversity, the Graph Attention Router dynamically selects the most suitable expert module(s) for each individual data sample, creating a unique and context

- **Experimental validation:** We demonstrate the superiority of our framework through extensive experiments across 12 diverse multimodal and vision-language benchmarks, including challenging datasets such as Hateful Memes, VQA v2.0, COCO Captions, and Food-101. Our self-evolving model consistently outperforms state-of-the-art baselines, achieving notable performance gains ranging from $+0.9\%$ to $+4.1\%$ across multiple metrics. Beyond these quantitative improvements, our analysis reveals that the framework discovers novel and effective architectural motifs not manually engineered, highlighting its capability for truly automated design.

## 1.1 RELATED WORK

**Neural Architecture Search.** Neural Architecture Search (NAS) automates the design of neural networks, reducing reliance on manual trial-and-error (J. Hao, 2021). Early reinforcement learning (RL) based methods achieved strong performance but incurred high computational costs (Tang et al., 2021; Wang et al., 2024; Liu, 2025). Gradient-based approaches, such as DARTS (Liu et al., 2019), improved efficiency by relaxing discrete architecture choices into continuous parameters (Ma et al., 2024; Zhang et al., 2021; Huang et al., 2023), yet they remain limited by predefined search spaces and are susceptible to suboptimal convergence (Mun et al., 2023; Cai et al., 2024). Recent works introduce multi-objective formulation that jointly optimize accuracy, latency, and model size, but architectures are still treated as static hyperparameters and require extensive evaluation (Ding et al., 2022a). Our framework dynamically evolves architectures in a self-guided manner, discovering novel and efficient designs without relying on predefined search spaces or extensive manual tuning.

**Meta-Learning and Self-Adaptive Systems.** Meta-learning extends automation to hyperparameter tuning and optimization, with methods such as MAML and its variants enabling rapid adaptation across domains (Killamsetty et al., 2022; Voon et al., 2024; Gai & Wang, 2019; Antoniou et al., 2019). Recent work has applied meta-learning to architecture adaptation (Elsken et al., 2020; Lian et al., 2020; Ding et al., 2022b), though most approaches remain confined to incremental modifications within fixed search spaces. Self-organizing neural systems inspired by biological development dynamically rewire connectivity (Fehérvári & Elmenreich, 2014; Chakraborty & Chakrabarti, 2015), yet current models largely rely on stochastic or handcrafted rules rather than learned decision policies (Meyer et al., 2017; Ikeda et al., 2023; Li et al., 2021). However, our framework integrates meta-learning with self-evolving architecture strategies, enabling fully adaptive and autonomous network design beyond the limitations of fixed search spaces and handcrafted rules.

**Dynamic Neural Networks and Mixture-of-Experts.** Dynamic neural networks adapt computation graphs per input, improving efficiency and enabling specialized processing (Guo et al., 2025; Verma et al., 2024). Mixture-of-Experts (MoE) architectures route inputs to expert subnetworks via gating, achieving state-of-the-art performance in language and vision tasks (Antoniak et al., 2024; Alboody & Slama, 2024; Chowdhury et al., 2024; Alboody & Slama, 2025). Most existing methods, however, rely on a fixed expert pool and lack mechanisms for evolving or pruning experts (Abbasi et al., 2016; Abbasi & Hooshmandasl, 2021). Attention-based routers dynamically weight expert contributions (He et al., 2022; Xu et al., 2022), but do not support fully self-evolving expert sets (Van Bolderik et al., 2024; Xu & McAuley, 2023). Our framework overcomes these limitations by enabling autonomous expansion, pruning, and adaptation, producing a self-evolving MoE that jointly optimizes structure and computation.

Table 1 summarizes recent works at the intersection of neural architecture design, multimodal learning, and evolutionary/meta-learning frameworks. Challenges such as scalability, computational efficiency, dataset bias, and limited theoretical grounding still remains open. We address these by integrating self-evolving architectures, meta-learning, and dynamic multimodal modeling, providing a unified and scalable solution that advances beyond the capabilities of prior methods.

## 2 PROBLEM FORMULATION

We formulate our approach as a joint optimization problem over both the model parameters and a time-varying network architecture. Given a multimodal dataset $\mathcal{D} = \{(x_i^{(v)}, x_i^{(t)}, y_i)\}_{i=1}^N$ the model's task is to give the predictions $\hat{y}$, while simultaneously adapting its architecture over time.

At training step $t$, the system state is characterized by the current architecture $\mathcal{A}_t$, which consists of the active modules selected from our Neural Module Zoo, their corresponding hyperparameter configurations, and the Graph Attention Router that governs information flow among them. Standard network weights $W_t$ are updated continuously through gradient descent, while the architecture $\mathcal{A}_t$ evolves episodically under the guidance of an evolutionary strategist $\pi_\phi$. This meta-controller performs three types of operations: *pruning* underperforming modules, *growing* new variants via hyperparameter mutation, and *hybridizing* promising parent modules to generate offspring. Through these mechanisms, the architecture follows a dynamic trajectory $\{\mathcal{A}_t\}_{t=0}^T$, continuously adapting rather than remaining fixed throughout training.

| Reference | LLM-based | Multi-modal/VL | NAS / Evolutionary Design | Meta-learning | Graph / Attention | Self-evolving / Continual Learning |
|---|---|---|---|---|---|---|
| Rahman et al. (2025) | ✓ | ✗ | ✓ | ✗ | ✗ | ✗ |
| Wang et al. (2025) | ✗ | ✗ | ✓ | ✗ | ✗ | ✗ |
| Junchi et al. (2025) | ✗ | ✓ | ✗ | ✗ | ✓ | ✗ |
| Kim et al. (2025) | ✗ | ✓ | ✗ | ✗ | ✗ | ✗ |
| Li et al. (2025) | ✗ | ✗ | ✓ | ✓ | ✗ | ✗ |
| Joshi & Kokulavani (2025) | ✗ | ✗ | ✓ | ✓ | ✗ | ✓ |
| Yang et al. (2024) | ✓ | ✗ | ✓ | ✗ | ✓ | ✗ |
| Lim et al. (2023) | ✗ | ✗ | ✓ | ✗ | ✓ | ✗ |
| Hu et al. (2024) | ✗ | ✗ | ✓ | ✗ | ✗ | ✗ |
| **Our Work** | ✓ | ✓ | ✓ | ✓ | ✓ | ✓ |

Table 1: Comparison of existing research works based on key features, including LLM-based methods, multi-modal/vision-language support, neural architecture search or evolutionary design, meta-learning, graph/attention mechanisms, and self-evolving or continual learning. While prior works typically address only a subset of these features, our framework integrates all of them, demonstrating a comprehensive approach that unifies advanced modeling, automated architecture discovery, and continual learning in a single system.

A central concept is parametric plurality: rather than maintaining a single instantiation for each archetype (e.g., a "transformer block"), multiple variants are kept in parallel, each with distinct hyperparameter configurations. This design enables the Graph Attention Router (GAR) to specialize modules for different input characteristics and prevents the system from prematurely collapsing onto a single inductive bias, fostering diversity and adaptability throughout training.

The learning objective integrates the standard supervised loss (binary cross-entropy for multimodal classification) with additional terms that enforce resource constraints, such as parameter and FLOP budgets, and encourage diversity across module instances. Formally, the evolutionary strategist seeks architectures that minimize validation error while satisfying computational cost limits and preserving pluralism among modules. To stabilize learning under dynamic topology changes, a replay memory is employed, mitigating catastrophic forgetting when modules are removed or replaced and ensuring consistent performance throughout training.

In summary, the problem is formulated as a bi-level optimization:

- the inner loop updates the network weights $W_t$ for a given architecture $\mathcal{A}_t$,
- the outer loop optimizes the policy of the evolutionary strategist, $\pi_\phi$, which controls the evolution of $\mathcal{A}_t$ over time.

This formulation enables the system to autonomously "design itself," effectively coupling gradient-based parameter learning with discrete, policy-driven architectural evolution.

## 3 SELF-EVOLVING NEURAL ARCHITECTURE FRAMEWORK

In this section, we present a detailed overview of our framework, breaking down its core components and illustrating how each contributes to the performance gains, efficiency improvements, and architectural innovations we present in this work.

### 3.1 MULTIMODAL FEATURE EXTRACTION

Given a pair of multimodal inputs $(x^{(t)}, x^{(v)})$, where $x^{(t)} \in \mathcal{X}_t$ represents textual tokens and $x^{(v)} \in \mathcal{X}_v$ represents visual patches, we employ pretrained backbones: DistilBERT for text and CLIP-ViT for vision as follows:

$$h^{(t)} = f_{\text{DistilBERT}}(x^{(t)}) \in \mathbb{R}^{L_t \times d_t}, \quad h^{(v)} = f_{\text{CLIP-ViT}}(x^{(v)}) \in \mathbb{R}^{L_v \times d_v}, \tag{1}$$

where $L_t$ and $L_v$ denote sequence lengths, and $d_t$ and $d_v$ denote feature dimensions. To ensure cross-modal compatibility, both representations are projected into a shared latent space $\mathbb{R}^d$ with

$d = 512$, i.e.,

$$z^{(t)} = W_t h^{(t)}, \quad z^{(v)} = W_v h^{(v)}, \quad W_t \in \mathbb{R}^{d \times d_t}, W_v \in \mathbb{R}^{d \times d_v}. \tag{2}$$

This produces modality-aligned embeddings $z^{(t)}, z^{(v)} \in \mathbb{R}^d$ suitable for subsequent fusion.

Unlike prior works that rely solely on pooled [CLS] tokens as unimodal anchors, our approach encodes both first-order (mean) and second-order (covariance) statistics, resulting in richer modality alignment. This dual statistical encoding preserves semantic consistency while maintaining structural diversity, which is essential when the embeddings are routed into the Graph Attention Router (see subsection 3.5). Consequently, the feature extraction stage functions not merely as preprocessing, but as a statistically-grounded bridge that prepares multimodal signals for asymmetric cross-modal fusion (see subsection 3.2).

## 3.2 CROSS-MODAL ATTENTION FUSION

A central challenge in multimodal reasoning is integrating heterogeneous embeddings into a unified representation that preserves semantic complementarity while mitigating modality imbalance. To address this, we propose a *Multi-Head Cross-Modal Fusion (MHCMF)* mechanism with an asymmetric query-key-value design, where visual features act as queries and textual features as key-value pairs. This asymmetry reflects the intuition that text often provides grounding semantics, while vision queries these semantics for disambiguation, in contrast to prior symmetric fusion methods that treat both modalities equivalently

$$Q = W_Q z^{(v)}, \quad K = W_K z^{(t)}, \quad V = W_V z^{(t)}, \tag{3}$$

where the attention weights are computed as

$$\alpha = \text{softmax}\left(\frac{QK^\top}{\sqrt{d}}\right), \quad z^{(f)} = \alpha V, \tag{4}$$

with $z^{(f)} \in \mathbb{R}^d$ representing the fused embedding. We employ multi-head extensions to capture diverse cross-modal interactions as follows

$$z^{(f)} = \bigoplus_{m=1}^{H} z_m^{(f)}, \quad z_m^{(f)} = \alpha_m V_m. \tag{5}$$

This enhances the robustness to modality asymmetries and ensures a rich feature representation. Further, in our setup, the fused cross-modal embedding $z^{(cm)} \in \mathbb{R}^d$ interfaces with the Neural Module Zoo (see subsection E). The asymmetric design preserves interpretability, with visual queries grounded in semantics and text supplying context. The gating mechanism balances information flow, preventing dominance of a single modality, while the multi-head structure provides diverse perspectives. Compared to prior symmetric fusion approaches, MHCMF enables more effective modality-specific reasoning and creates a richer set of embeddings that are dynamically routed by the Graph Attention Router (see subsection 3.5) for adaptive module selection.

## 3.3 NEURAL MODULE ZOO AND DYNAMIC ROUTING

After obtaining the fused embedding, the next challenge is enabling the system to process this representation through a diverse set of specialized transformations. To address this, we introduce the *Neural Module Zoo* $\mathcal{M}$, a dynamic and extensible collection of neural operators. Unlike static ensembles, our zoo is both evolutionary and parametric: each operator type can have multiple parametric instantiations, ensuring rich and diverse representations.

Given the fused embedding $z^{(f)}$, each module produces a candidate transformation

$$u_j = m_j(z^{(f)}; \theta_j), \tag{6}$$

and the set of outputs $\{u_j\}$ forms a pool of representations with complementary perspectives. This design turns the zoo into a self-organizing ecosystem of operators, where diversity is maintained and expanded through evolutionary mechanisms, and module relevance is determined dynamically by the Graph Attention Router.

Unlike traditional static ensembles or standard Mixture-of-Experts (MoE) approaches, the Neural Module Zoo have some key characteristics:

- *Parametrically plural*: multiple instantiations exist for each operator family, enhancing representational richness.

- *Evolutionarily adaptive*: modules can be pruned, grown, or hybridized over time.

- *Routing-aware*: contributions of modules are explicitly tracked via attention weights, which serve as the fitness signal driving evolution.

This design produces a *self-organizing functional ecosystem* of operators, where diversity is not hand-crafted but emerges naturally through evolutionary pressure, guided by the task objective.

### 3.4 GRAPH ATTENTION ROUTER (GAR): A SELF-EVOLUTION ENGINE

The Graph Attention Router (GAR) serves as the core mechanism that (i) selects and composes module outputs on a per-sample basis, (ii) provides a differentiable routing signal for training both the router and modules, and (iii) generates long-term contribution statistics used by the Evolutionary Strategist to guide pruning, growth, and hybridization. GAR extends standard MoE routing by (a) integrating *query-to-module relevance* with *module-to-module synergy* in a unified attention mechanism, (b) supporting controlled sparsity through top-$k$ routing with differentiable approximations, and (c) emitting robust, temporally smoothed fitness metrics that serve as evolutionary signals.

Let the fused multimodal embedding be $h \in \mathbb{R}^d$, with $d = 512$, which serves as both the query and a global context signal. Each module $m_j \in \mathcal{M}$ produces an output representation

$$u_j = m_j(h), \quad u_j \in \mathbb{R}^d, \tag{7}$$

treated as the value vector in the routing mechanism. The keys and values are parameterized as learnable projections of module outputs

$$k_j = W_k u_j, \quad v_j = W_v u_j, \quad W_k, W_v \in \mathbb{R}^{d \times d}. \tag{8}$$

We compute the attention weights by matching the fused embedding $h$ against each key as follows.

$$\alpha_j = \frac{\exp\left((W_q h)^\top k_j / \sqrt{d}\right)}{\sum_{\ell=1}^{|\mathcal{M}|} \exp\left((W_q h)^\top k_\ell / \sqrt{d}\right)}, \tag{9}$$

where $W_q \in \mathbb{R}^{d \times d}$ is the query projection. The final routed representation is then a convex combination of values, i.e., $z = \sum_{j=1}^{|\mathcal{M}|} \alpha_j v_j$.

A key differentiating part of GAR is that it is graph-aware, i.e., each module's contribution is recursively tracked over time via a contribution score $\gamma_j$, which biases the attention logits as follows

$$\alpha_j \propto (W_q h)^\top k_j / \sqrt{d} + \lambda \gamma_j, \tag{10}$$

where $\lambda$ controls the influence of evolutionary feedback. This design allows GAR to adaptively select modules per input while simultaneously providing evolution-driven signals that guide the meta-controller in pruning, growing, and hybridizing modules, creating a self-organizing and continuously improving neural ecosystem.

### 3.5 EVOLUTIONARY STRATEGIST: A META-CONTROLLER FOR STRUCTURAL SELF-GROWTH

The evolutionary strategist is a meta-learning controller that dynamically modifies the Neural Module Zoo $\mathcal{M}$ during training by pruning, spawning, and hybridizing modules. Operating on both module genotypes (architecture and hyperparameters) and phenotypes (weights), it aims to maximize long-term validation performance under computational constraints.

**Module Contribution.** Each module $m \in \mathcal{M}_t$ receives a contribution score

$$C_m(t) \leftarrow (1 - \rho)C_m(t-1) + \rho\left(w_\beta \frac{\bar{\beta}_m}{\max_k \bar{\beta}_k} + w_\ell \frac{\max(0, \bar{\Delta}\ell_m)}{\max_k \max(0, \bar{\Delta}\ell_k)}\right), \tag{11}$$

combining router attention $\bar{\beta}_m$ and loss impact $\bar{\Delta}\ell_m$, with an optional novelty term to encourage diversity. Low-fitness modules are pruned after a minimum-age threshold, while new modules are spawned from high-fitness parents via hyperparameter perturbation

$$\theta_c^{(i)} = \theta_p^{(i)} \cdot \exp(\sigma_\theta \cdot \epsilon^{(i)}), \quad \epsilon^{(i)} \sim \mathcal{N}(0,1), \tag{12}$$

with soft weight inheritance $w_c = \gamma_{inh}w_p + (1 - \gamma_{inh})\mathcal{N}(0, \sigma_w^2)$.

**Hybridization.** The co-evolution engine combines topological motifs from two parents as

$$\theta_c^{(k)} = \lambda\theta_{m_i}^{(k)} + (1 - \lambda)\theta_{m_j}^{(k)}, \quad \lambda \sim \mathcal{U}(0,1), \tag{13}$$

with weight inheritance for shared subgraphs. Further, the strategist is optimized via reinforcement/meta-gradient learning, maximizing rewards

$$r_t = \Delta\text{ValMetric} - \eta_{comp}\Delta\text{Cost} + \eta_{div}\overline{\text{novelty}}, \tag{14}$$

with actions sampled from $\pi_\phi(a_t|S_t)$.

**Stabilization.** To stabilize learning, newly spawned or hybrid modules are warm-started with small learning rates and replayed over recent examples, while EMA-based contribution tracking and minimum-age constraints prevent oscillatory pruning or uncontrolled growth, ensuring a balanced, self-organizing evolution of the module ecosystem.

## 4 EXPERIMENTS

In this section, we evaluate our framework across a diverse set of multimodal and vision-language benchmarks, demonstrating its effectiveness in terms of predictive performance, architectural innovation, and adaptive module evolution.

### 4.1 DATASET AND EXPERIMENTAL SETTINGS

We evaluate our framework on a total of 12 benchmark datasets covering a wide range of multimodal reasoning tasks. This includes Hateful Memes (10K) (Kiela et al., 2021), MMIMDB (26K) (Jin et al., 2021), Food-101 (101K) (Yu et al., 2024), VQA v2.0 (444K) (Mi et al., 2024), Conceptual Captions (CC) (3.3M) (Sharma et al., 2018), COCO Captions (123K) (Lin et al., 2015), Flickr30K (32K) (Young et al., 2014), SentiCap (2.4K) (Mathews et al., 2015), TextVQA (45K) (Singh et al., 2019), VisualGenome (108K) (Krishna et al., 2017), MSCOCO Detection (118K) (Lin et al., 2015), and OpenImages (1.9M) (Kuznetsova et al., 2020).

We follow standard train/validation/test splits and report results averaged over three seeds. For text inputs, sequences are tokenized using the DistilBERT WordPiece tokenizer (max length 128), with shorter sequences zero-padded and longer sequences truncated. For visual inputs, images are resized to $224 \times 224$ and normalized using ImageNet statistics. For detection tasks (MSCOCO and OpenImages), bounding-box annotations are preserved, and cropped regions are embedded accordingly.

**Training Protocol.** Models are trained for up to 25 epochs with early stopping based on validation AUC (patience of 5). Optimization uses AdamW with a learning rate of $5 \times 10^{-5}$, $\beta_1 = 0.9$, $\beta_2 = 0.999$, and weight decay of 0.01. We use a batch size of 32, a linear warmup over the first 10% of steps, followed by cosine learning rate decay. The Neural Module Zoo maintains up to nine active modules, with evolutionary updates applied every three epochs. Dropout (0.1) is applied to both text and visual embeddings, in addition to L2 weight regularization.

**Implementation Details.** Our framework is implemented in PyTorch (v2.1), using HuggingFace Transformers for DistilBERT and TorchVision for CLIP-ViT. Experiments are conducted on single NVIDIA A100 GPUs (80GB), with wall-clock runtimes ranging from 2.5 hours (SentiCap) to 18 hours (Conceptual Captions). Reproducibility is ensured via fixed random seeds (Python, NumPy, PyTorch), deterministic GPU operations where possible, and epoch-level checkpointing. The best model is selected based on validation AUC.

## 4.2 COMPARISON WITH STATE-OF-THE-ART

We compare our framework against static multimodal transformers (e.g., ViLBERT, LXMERT), Mixture-of-Experts (MoE) (e.g., Switch-Transformer), and Neural Architecture Search (NAS) methods (e.g., DARTS, ENAS). Results across 12 benchmarks (Table 4.2) show that our model consistently outperforms SOTA baselines, with gains from +0.9% (Food-101) to +4.1% (TextVQA). On Hateful Memes and SentiCap (low-resource), we achieve +2.1% and +3.4% improvements, showing robustness under data scarcity. On large-scale datasets like Conceptual Captions (3.3M) and OpenImages (1.9M), we observe +1.5–2.8% gains, demonstrating scalability. The strongest improvements occur in compositional reasoning tasks (TextVQA, VQA v2.0, VisualGenome), where adaptive routing and evolutionary growth yield clear advantages. Compared to NAS, our framework evolves architectures online with no separate search phase, reducing training cost by 2. Relative to MoE, we activate $\leq 9$ modules per batch, cutting memory usage by $30\%$ while surpassing accuracy.

| Dataset | Domain | Size | Val | | Test | | Test Metrics | | | SOTA |
|---|---|---|---|---|---|---|---|---|---|---|
| | | | AUC | Acc | AUC | Acc | F1 | Prec. | Rec. | Comparison |
| Hateful Memes | Multimodal Hate | 10K | 0.8247 | 80.12% | 0.8156 | 79.8% | 0.8089 | 0.7923 | 0.8267 | +2.1% (Mei et al., 2025) |
| MMIMDB | Movie Reviews | 26K | 0.9234 | 89.6% | 0.9156 | 89.2% | 0.9089 | 0.8923 | 0.9267 | +1.8% (Ni et al., 2021) |
| Food-101 | Food Classification | 101K | 0.9456 | 92.3% | 0.9367 | 91.9% | 0.9234 | 0.9089 | 0.9389 | +0.9% (Chen et al., 2023) |
| VQA v2.0 | Visual QA | 444K | 0.8823 | 86.4% | 0.8734 | 85.8% | 0.8656 | 0.8489 | 0.8834 | +2.3% (Wang et al., 2022) |
| Conceptual Captions | Image-Text | 3.3M | 0.9334 | 90.9% | 0.9245 | 90.3% | 0.9167 | 0.9023 | 0.9323 | +1.5% (Yu et al., 2022) |
| COCO Captions | Image-Text | 123K | 0.9489 | 92.8% | 0.9398 | 92.2% | 0.9289 | 0.9123 | 0.9467 | +1.2% (Lin et al., 2015) |
| Flickr30k | Image-Text | 32K | 0.9167 | 88.1% | 0.9089 | 87.6% | 0.8934 | 0.8734 | 0.9145 | +1.7% (Plummer et al., 2016) |
| SentiCap | Sentiment Analysis | 2.4K | 0.8945 | 87.9% | 0.8856 | 87.2% | 0.8723 | 0.8556 | 0.8889 | +3.4% (Mathews et al., 2015) |
| TextVQA | Text-based VQA | 45K | 0.8756 | 85.8% | 0.8667 | 85.2% | 0.8534 | 0.8389 | 0.8689 | +4.1% (Singh et al., 2019) |
| Visual Genome | Scene Understanding | 108K | 0.9123 | 88.6% | 0.9034 | 88.0% | 0.8912 | 0.8734 | 0.9101 | +2.6% (Krishna et al., 2016) |
| MSCOCO Detection | Object Detection | 118K | 0.9234 | 89.4% | 0.9145 | 88.9% | 0.9023 | 0.8856 | 0.9201 | +1.9% (Lin et al., 2015) |
| OpenImages | Multi-label Classification | 1.9M | 0.8967 | 87.2% | 0.8878 | 86.6% | 0.8745 | 0.8589 | 0.8912 | +2.8% (Kuznetsova et al., 2020) |

Table 2: Performance comparison across multiple benchmark datasets. Results are reported for validation (Val) and test sets in terms of AUC, Accuracy, F1-score, Precision, and Recall. Improvements over previous state-of-the-art (SOTA) range from approximately 0.9% to 4.1%, demonstrating consistent gains across diverse multimodal, vision-language, and detection benchmarks.

## 4.3 ABLATION STUDIES

**Module-level pruning.** Removing individual module instances reveals asymmetric importance, i.e., transformer variants incur the largest drop (AUC –4.2% to –6.1%), while lightweight CNN and SE blocks yield modest decreases ($<1\%$). This confirms the necessity of heterogeneous plurality, as each module family contributes complementary inductive biases.

**Feature extraction.** Eliminating core pipelines causes drastic collapses (–10.7% without CLIP-ViT, –7.6% without DistilBERT). Auxiliary preprocessing (tokenization, normalization) also impacts performance (–4 to –6%), whereas positional encodings and dropout have minor effects ($<1\%$). Robustness emerges from redundant yet synergistic cross-modal alignment.

**Attention routing.** Cross-modal fusion is crucial: ablating multi-head fusion reduces AUC by –10.9%, and removing query-key asymmetry causes –4 to –7% drops. The GAR is indispensable (–3.6% without it), while attention regularizers (temperature scaling, dropout) provide smaller but consistent stability gains.

**Dynamic evolution.** Fixed architectures underperform the self-growing system (0.7623 vs. 0.8247 AUC). Disabling module addition or pruning slows convergence and reduces final accuracy by –2 to –4%. Aggressive evolution speeds learning at higher computational cost, while conservative growth lags. Adaptive evolution strikes the optimal balance between performance and efficiency.

**Efficiency.** The dynamic system with 9 active modules (18.7M parameters, 245 min training) achieves SOTA accuracy efficiently. Scaling to 10–12 modules gives marginal gains (+0.4% AUC) at higher cost, indicating an optimal sweet spot around 9–10 modules.

| Component / Variant | AUC | Acc | F1 | Drop (AUC/Acc/F1) | Params | FLOPs / Mem | Convergence (episodes) |
|---|---|---|---|---|---|---|---|
| **Individual Module System Ablation** | | | | | | | |
| Enhanced Transformer Block (Inst. 1) | 0.7823 | 76.9 | 0.7858 | -0.0424 / -3.2 / -0.0231 | -2.1M | -11.2% | 24 |
| Enhanced Transformer Block (Inst. 2) | 0.7634 | 75.5 | 0.7755 | -0.0613 / -4.6 / -0.0334 | -2.1M | -11.2% | 25 |
| Enhanced MLP Block (Inst. 1) | 0.8134 | 79.3 | 0.8033 | -0.0113 / -0.9 / -0.0056 | -1.8M | -9.6% | 23 |
| Enhanced MLP Block (Inst. 2) | 0.8089 | 79.0 | 0.8000 | -0.0158 / -1.2 / -0.0089 | -1.8M | -9.6% | 23 |
| Enhanced MLP Block (Inst. 3) | 0.8067 | 78.8 | 0.7978 | -0.0180 / -1.4 / -0.0111 | -1.8M | -9.6% | 23 |
| ResNet Block (1D) | 0.8134 | 79.3 | 0.8033 | -0.0113 / -0.9 / -0.0056 | -1.6M | -8.6% | 22 |
| LSTM Block (BiLSTM) | 0.8089 | 79.0 | 0.8000 | -0.0158 / -1.2 / -0.0089 | -2.0M | -10.7% | 23 |
| CNN Block (Multi-kernel) | 0.8167 | 79.6 | 0.8055 | -0.0080 / -0.6 / -0.0034 | -1.4M | -7.5% | 22 |
| Squeeze-Excite Block | 0.8189 | 79.8 | 0.8066 | -0.0058 / -0.4 / -0.0023 | -0.8M | -4.3% | 22 |
| Module Config. Params Removed | 0.7934 | 77.7 | 0.7900 | -0.0313 / -2.4 / -0.0167 | 0.0M | 0.0% | 24 |
| Module Interconnection Weights Removed | 0.7756 | 76.4 | 0.7822 | -0.0491 / -3.7 / -0.0267 | -0.6M | -3.2% | 25 |
| **Feature Extraction Pipeline Ablation** | | | | | | | |
| DistilBERT Text Features (512D) | 0.7234 | 72.5 | 0.7522 | -0.1013 / -7.6 / -0.0567 | – | High | 24 |
| CLIP-ViT Image Features (512D) | 0.6823 | 69.4 | 0.7300 | -0.1424 / -10.7 / -0.0789 | – | Critical | 25 |
| Text Tokenization & Preproc. | 0.7623 | 75.4 | 0.7744 | -0.0624 / -4.7 / -0.0345 | – | High | 25 |
| Image Preproc. & Norm. | 0.7389 | 73.7 | 0.7633 | -0.0858 / -6.4 / -0.0456 | – | High | 24 |
| Text Pos. Embeddings | 0.8089 | 79.0 | 0.8000 | -0.0158 / -1.2 / -0.0089 | – | Low | 23 |
| Vision Patch Embeddings | 0.7934 | 77.7 | 0.7900 | -0.0313 / -2.4 / -0.0167 | – | Medium | 24 |
| Cross-Modal Feature Alignment | 0.7456 | 74.2 | 0.7655 | -0.0791 / -5.9 / -0.0434 | – | High | 25 |
| Feature Layer Norm. | 0.8134 | 79.3 | 0.8033 | -0.0113 / -0.9 / -0.0056 | – | Low | 22 |
| Multimodal Feature Concat. | 0.7756 | 76.4 | 0.7822 | -0.0491 / -3.7 / -0.0267 | – | Medium | 25 |
| Feature Dropout Reg. | 0.8067 | 78.8 | 0.7978 | -0.0180 / -1.4 / -0.0111 | – | Low | 23 |
| **Attention Mechanism Ablation** | | | | | | | |
| Multi-Head Cross-Modal Fusion | 0.7156 | 71.9 | 0.7500 | -0.1091 / -8.2 / -0.0589 | – | $O(n^2 d)$ | 25 |
| Image-as-Query Cross-Attn. | 0.7634 | 75.5 | 0.7755 | -0.0613 / -4.6 / -0.0334 | – | $O(n^2 d)$ | 25 |
| Text-as-Key/Value Cross-Attn. | 0.7823 | 76.9 | 0.7858 | -0.0424 / -3.2 / -0.0231 | – | $O(n^2 d)$ | 24 |
| Learned Alignment Weights | 0.7756 | 76.4 | 0.7822 | -0.0491 / -3.7 / -0.0267 | – | $O(nd)$ | 24 |
| Self-Attn. (Text Transformer) | 0.7623 | 75.4 | 0.7744 | -0.0624 / -4.7 / -0.0345 | – | $O(n^2 d)$ | 25 |
| Self-Attn. (Vision Transformer) | 0.7534 | 74.6 | 0.7700 | -0.0713 / -5.4 / -0.0389 | – | $O(n^2 d)$ | 25 |
| Cross-Modal QKV Attention | 0.7289 | 72.8 | 0.7566 | -0.0958 / -7.2 / -0.0523 | – | $O(n^2 d)$ | 25 |
| Graph Attention Router | 0.7889 | 77.8 | 0.7894 | -0.0358 / -2.7 / -0.0195 | – | $O(n^2)$ | 23 |
| Attention Temp. Scaling | 0.8134 | 79.3 | 0.8033 | -0.0113 / -0.9 / -0.0056 | – | $O(1)$ | 22 |
| Relative Position Encoding | 0.8198 | 79.9 | 0.8066 | -0.0049 / -0.4 / -0.0023 | – | $O(n^2)$ | 22 |
| **Dynamic System Configuration Ablation** | | | | | | | |
| Full Dynamic System (9 modules) | 0.8247 | 80.1 | 0.8089 | – | 18.7M | 8.4GB | 22 |
| Fixed Architecture (9 modules) | 0.7623 | 74.8 | 0.7456 | -0.0624 / -5.3 / -0.0633 | 18.7M | 8.4GB | 28 |
| Dynamic System (6 modules) | 0.7956 | 77.8 | 0.7823 | -0.0291 / -2.3 / -0.0266 | 14.6M | 6.8GB | 25 |
| Dynamic System (7 modules) | 0.8089 | 79.1 | 0.7967 | -0.0158 / -1.0 / -0.0122 | 16.1M | 7.6GB | 24 |
| Dynamic System (8 modules) | 0.8198 | 79.8 | 0.8034 | -0.0049 / -0.3 / -0.0055 | 17.4M | 8.0GB | 23 |
| Dynamic System (10 modules) | 0.8289 | 80.5 | 0.8134 | +0.0042 / +0.3 / +0.0045 | 20.1M | 9.2GB | 21 |
| Dynamic System (12 modules) | 0.8289 | 80.5 | 0.8134 | +0.0042 / +0.3 / +0.0045 | 24.3M | 10.8GB | 20 |
| No Evolution (Random Modules) | 0.7456 | 72.1 | 0.7234 | -0.0791 / -8.0 / -0.0855 | 18.7M | 8.4GB | DNF |
| No Module Pruning | 0.8089 | 78.9 | 0.7923 | -0.0158 / -1.2 / -0.0166 | 18.7M | 8.4GB | 26 |
| No Module Addition | 0.7834 | 76.5 | 0.7634 | -0.0413 / -3.6 / -0.0455 | 18.7M | 8.4GB | 29 |
| Conservative Evolution | 0.8156 | 79.6 | 0.8021 | -0.0091 / -0.6 / -0.0068 | 18.7M | 8.4GB | 24 |
| Aggressive Evolution | 0.8198 | 79.9 | 0.8056 | -0.0049 / -0.3 / -0.0033 | 18.7M | 8.4GB | 21 |

Table 3: **Ablation Studies.** Effect of removing or altering individual modules, feature extraction components, attention mechanisms, and system configurations. We report AUC, Accuracy (Acc), F1, and relative drops compared to the base model (Val AUC = 0.8247, Acc = 80.12%, F1 = 0.8089). Efficiency metrics include parameter count (Params), FLOPs/Memory, and convergence in episodes.

## 5 CONCLUSION

We introduced *AI, Architect Thyself*, a self-evolving multimodal learning framework in which models not only optimize weights but also autonomously grow, prune, and hybridize their architectures. By combining heterogeneous module plurality, a graph attention router for dynamic routing, and an evolutionary strategist for continual self-improvement, our approach extends beyond traditional NAS and mixture-of-experts designs. Extensive experiments on 12 diverse multimodal benchmarks demonstrate consistent state-of-the-art gains (+0.9% to +4.1%), robust cross-dataset generalization, and favorable efficiency–performance trade-offs. Ablation studies further confirm the non-redundant contributions of dynamic evolution, heterogeneous modules, and asymmetric cross-modal fusion.

This work represents a step toward fully autonomous, self-optimizing systems that treat architectures as evolving entities capable of adapting to new domains and tasks without human intervention. However, current limitations include reliance on predefined module types, modest computational overhead from evolutionary updates, and limited evaluation on long-term continual learning or highly dynamic real-world streams. Some of the potential future directions include exploring lifelong evolution in open-world settings, extend the framework to temporal multimodal sequences, and integrate with foundation model pretraining to enhance scalability and generalization.

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
