APPENDIX

## A   DIFFERENCE BETWEEN TRADITIONAL NAS AND MALS

Traditional NAS iterates in outer loops, fully retraining candidate architectures between searches. Our proposed framework MALS here interleaves architectural updates with gradient updates using

- Micro-timescale ($\tau_g$): Standard stochastic gradient descent updates the model weights.
- Macro-timescale ($\tau_a$): Every $k$ gradient steps, the Meta Controller executes an evolutionary adaptation event.

If $\tau_a \ll \tau_g$, the architecture can quickly adapt to novel data patterns without overfitting stale topologies. This dual time-scale formalism can be expressed as in Eq. 15.

$$\theta_{t+1} = \theta_t - \eta_g \nabla_\theta \mathcal{L}_{task}(\theta_t, \mathcal{M}_t)$$
$$\mathcal{M}_{t+1} = \mathcal{F}_{evolve}(\mathcal{M}_t, \pi_\theta, \mathcal{H}_t) \quad \text{only if} \quad t \mod k = 0 \tag{15}$$

Here, $\theta$ represents module parameters, and $\mathcal{F}_{evolve}$ is the learned evolutionary update function.

## B   PROBLEM FORMULATION AND ARCHITECTURAL OVERVIEW

### B.1   NOTATION AND CORE OBJECTS

Let

- $\mathcal{D} = \{(x_i^{(v)}, x_i^{(t)}, y_i)\}_{i=1}^N$ be the dataset of multimodal examples (visual, textual, label), drawn i.i.d. from an unknown distribution $\mathcal{P}_{data}$.
- $d$ be the shared latent dimension (we use $d = 512$ in experiments).
- $\mathcal{A}_t$ denotes the **architecture state** at training step (or epoch) $t$. $\mathcal{A}_t$ comprises:
    - a set of active modules (the **Neural Module Zoo**) $\mathcal{M}_t = \{m_{t,1}, \ldots, m_{t,N_t}\}$,
    - router parameters $\theta_t^{(r)}$,
    - module hyperparameter descriptors $\Theta_t = \{\theta_{t,1}, \ldots, \theta_{t,1}\}$ (these describe structural choices like layers, heads, droupout, activation type),
    - global resource counters (parameter count, FLOPs).
- $W_t$ denotes all learnable weights at step $t$: module weights, router weights, projection heads, classifier head, and any meta-controller weights (except where separated explicitly).
- $\pi_\phi$ denote the **Evolutionary Strategist** (meta-controller) parameterized by $\phi$; it issues discrete/continuous actions that transform $\mathcal{A}_t \mapsto \mathcal{A}_{t+1}$/

A single forward pass on sample $x$ under architecture $\mathcal{A}_t$ yields prediction $\hat{y}(x : W_t, \mathcal{A}_t)$. The per-sample task loss is $l(\hat{y}, y)$, e.g. cross-entropy.

**Why this representation?**   Treating architecture as an explicit, time-indexed object $\mathcal{A}_t$ makes it possible to 1) reason about changes over training, 2) define budget constraints that vary over time, and 3) expose $\pi_\phi$ a state on which to condition actions — all necessary for principled co-evolution.

### B.2   JOINT (BI-LEVEL) OPTIMIZATION: WEIGHTS AND TOPOLOGY

We designed this as a bi-level optimization where weights are optimized continuously while the meta-controller optimizes the architecture trajectory:

$$\begin{aligned} \text{(Outer / meta)} \quad & \min_\phi \mathbb{E}[\mathcal{L}_{val}(W_T(\phi), \mathcal{A}_T(\phi))] \\ \text{subject to} \quad & \mathcal{A}_{t+1} \sim \pi_\phi(\cdot|s_t), t = 0, \ldots, T-1, \\ \text{(Inner/ weights)} \quad & W_{t+1} = \mathcal{U}(W_t, \nabla_{W_t} \mathcal{L}_{train}(W_t, \mathcal{A}_t)), \end{aligned} \tag{16}$$

where:

- $\mathcal{L}_{train}$ and $\mathcal{L}_{val}$ are empirical train and validation losses,
- $\mathcal{U}$ denotes the inner-loop optimizer (SGD/Adam step),
- $s_t$ is the strategist state,
- expectations are over data sampling and any stochastic components of $\pi_\phi$.

**Why a bi-level view?** Architecture decisions change the downstream loss landscape; optimizing $\phi$ requires evaluating the effect of architectural actions after weight updates. The bi-level view captures this causal dependency. Directly solving this exact bi-level problem is computationally intractable for large models, so we adopt approximations (meta-gradient, reward shaping, and fitness proxies) discussed below.

## B.3 PARAMETRIC PLURALITY: CONFIGURATION SPACES AND MODULE INSTANCING

We define an **archetype set** $\mathcal{T}$ (e.g., Transformer, LSTM, ResNet, MLP, Squeeze-Excite). For each archetype $a \in \mathcal{T}$, we defined a configuration (hyperparameter) space $\Omega_a$. A module instance is then:

$$m = (a, \theta^{(arch)}, \omega), \quad \theta^{(arch)} \in \Omega_a, \omega = \text{learned weights} \tag{17}$$

We denoted the probability distribution over configurations as $P(\theta^{(arch)}|a)$ - the strategist can sample from or choose points in this space.

**Parametric plurality** means for a fixed archetype $a$, we allow multiple instances $\{m_i\}$ with different $\theta_i^{(arch)}$. Formally:

$$\mathcal{M}_t = \bigcup_{a \in \mathcal{T}} \{m_{t,i}^{(a)} : \theta_{t,i}^{(arch)} \sim P_t(\cdot|a)\}. \tag{18}$$

**Why?** Because of two main reasons, the first one being that multiple instantiations of the same structural bias with different internal hyperparameters produce distinct inductive priors and optimization dynamics. The second one is reducing reliance on a single optimum configuration for an archetype and enables per-sample specialization via the router.

Here, we quantify module diversity with a metric $\mathcal{D}(\mathcal{M}_t)$,

$$\mathcal{D}(\mathcal{M}_t) = \frac{1}{N_t^2} \sum_{i,j} \Delta(\theta_{t,i}^{(arch)}, \theta_{t,j}^{(arch)}) + \frac{1}{N_t^2} \sum_{i,j} \mathbb{E}_x ||u_{t,i}(x) - u_{t,j}(x)||_2, \tag{19}$$

where $\Delta$ measures configuration distance (mixed categorical/continuous) and the second term measures output diversity.

## B.4 ROUTER, CONTRIBUTION, AND THE STRATEGIST STATE

The Graph Attention Router (GAR) produces a per-sample distribution over modules:

$$\alpha(x; \mathcal{A}_t, W_t) = \text{GAR}(f(x; \mathcal{A}_t, W_t), \mathcal{M}_t) \in \Delta^{N_t - 1}, \tag{20}$$

and routed representation $z^{(r)} = \sum_m \alpha_m v_m$. To make an evolution decision, the strategist receives summary statistics (the **state** $s_t$) that include per-module fitness traces $\Phi_{t,m}$ (defined in 3.5), module utilization $\bar{\alpha}_{t,m}$, resource vector $c(\mathcal{A}_t)$ (parameter count, FLOPs, latency), global performance indicators and diversity $\mathcal{D}(\mathcal{M}_t)$.

**Why these state features?** They connect short-term routing behavior (utilization) with long-term utility (fitness), and expose resource constraints so $\pi_\phi$ can make capacity-aware decisions (prune low-utility modules, grow when capacity allows).

## B.5 EVOLUTIONARY OPERATORS

The strategist operates via a small set of operators that map architectures to architectures:

- **Prune operator** $\mathcal{P}_\tau$: We remove modules $m$ with $\Phi_{t,m} < \tau_{prune}$ for $T_{patience}$ steps.

- **Mutate/Grow operator** $\mathcal{G}$: We sampled a parent $m_p$ (probability proportional to positive fitness) and create child $m_c$ by:

$$\theta_c^{(arch)} = \theta_p^{(arch)} + \epsilon, \quad \epsilon \sim \mathcal{N}(0, \sigma_{mut}^2), \tag{21}$$

  and initialize weights $\omega_c$ (either random or derived via partial weight inheritance).

- **Hybridization/Crossover operator** $\mathcal{H}$: For parents $m_i, m_j$, we selected proportional to fitness, and produced a child with mixed hyperparameters:

$$\theta_c^{(arch)} = \text{CROSS}(\theta_i^{(arch)}, \theta_j^{(arch)}), \tag{22}$$

  where CROSS handles continuous parameters by convex combination and categorical parameters by probabilistic selection or learned mapping (e.g., one parent chosen per categorical field with probability proportional to fitness).

- **Reinsertion/Assignment:** The newly created modules are inserted into $\mathcal{M}_t$ if resource budget permits, else they replace low-fitness modules.

The selection probabilities for parents are softmaxed fitness scores:

$$P(m_i, \text{chosen}) = \frac{\exp(\Phi_{t,i}/\tau_{sel})}{\sum_j \exp(\Phi_{t,i}/\tau_{sel})}. \tag{23}$$

**Why these operators?** They emulate biological mechanisms while remaining interpretable and tunable. Crossover blends complementary traits; mutation explores local neighbourhoods; pruning removes dead weight. Soft selection and patience thresholds prevent noisy immediate deletions.

## B.6 CONSTRAINTS AND RESOURCE-AWARE OBJECTIVE

Real systems operate under budgets. Let $C(\mathcal{A}_t)$ be a vector of costs (parameters, inference latency per sample, memory). The strategist must respect constraints $C(\mathcal{A}_t) \preceq C_{max}$. We embedded resource costs into the meta reward so the strategist optimizes utility under budgets. We defined the per-decision reward (to be maximized):

$$r_t = -\mathcal{L}_{val}(W_t, \mathcal{A}_t) - \lambda_c \cdot \text{cost}(C(\mathcal{A}_t)) + \lambda_d \mathcal{D}(\mathcal{M}_t), \tag{24}$$

where $\text{cost}(\cdot)$ aggregates resource usage into a scalar penalty and $\mathcal{D}(\cdot)$ is the diversity reward. The outer optimization becomes:

$$\max_\phi \mathbb{E}_{\pi_\phi} \left[ \sum_{t=0}^{T-1} \gamma_{disc}^t r_t \right]. \tag{25}$$

**Why reward shaping?** Directly minimizing final validation loss is costly to estimate. A dense reward combining validation performance, resource penalties, and diversity fosters architectures that generalize, are efficient, and preserve pluralism.

**Replay memory and stability** Architecture changes introduce non-stationarity. To stabilize training, we maintain a replay buffer $\mathcal{R}$ storing representative samples (and their labels). When a module changes (spawned, hybridized), we interleave replay training on $\mathcal{R}$ to preserve past capabilities:

$$W_{t+1} \leftarrow \mathcal{U}(w_t, \nabla_{W_t}[\mathcal{L}_{train}(W_t, \mathcal{B}) + \mu \mathcal{L}_{replay}(W_t, \mathcal{R})]). \tag{26}$$

This is important as replay mitigates catastrophic forgetting when architecture topology changes and modules are inserted/removed. It also provides a stable baseline for computing module utility.

### B.7 PRACTICAL APPROXIMATIONS AND ALGORITHMIC SUMMARY

Solving the exact bi-level is impractical. Therefore, we adopted these approximations:

1. **Local fitness proxies:** We used $\Phi_{t,m}$ instead of full retraining-based evaluation for parent selection.

2. **Policy optimization:** We trained $\pi_\phi$ with reinforcement learning (PPO/actor-critic) using the dense reward $r_t$.

3. **Warm-start and patience:** We delayed pruning/hybridization for $E_{warm}$ epochs to allow modules and router to stabilize.

4. **Deterministic operations at eval time:** We sparsified via deterministic top-K for reproducible inference.

**Algorithmically.** We alternated inner-loop updates of $W_t$ (with replay) with occasional strategist decision steps that apply $\mathcal{P}, \mathcal{G}, \mathcal{H}$ based on $\Phi$ and $s_t$. The GAR provides per-sample routing and the contribution traces that ground evolutionary choices.

## C MULTIMODAL FEATURE EXTRACTION

Let the input pair be $(x^{(t)}, x^{(v)})$, where $x^{(t)} \in \mathcal{X}$, denotes a sequence of text tokens and $x^{(v)} \in \mathcal{X}_v$ denotes an image decomposed into visual patches. Our objective is to map these heterogeneous modalities into a shared latent manifold $\mathcal{Z} \subseteq \mathbb{R}^d$, enabling subsequent cross-modal alignment and adaptive modular routing.

### C.1 TEXTUAL ENCODING

We tokenize the text sequence as

$$x^{(t)} = \{w_1, w_2, \ldots, w_{L_t}\}, \quad w_i \in \mathcal{V}, \tag{27}$$

where $\mathcal{V}$ is the vocabulary. A pretrained DistilBERT encoder $f_t : \mathcal{X}_t \in \mathbb{R}^{L_t \times d_t}$ produces contextualized embeddings:

$$h^{(t)} = f_t(x^{(t)}), \quad h^{(t)} = [h_1^{(t)}, h_2^{(t)}, \ldots, h_{L_t}^{(t)}], \quad h_i^{(t)} \in \mathbb{R}^{d_t}. \tag{28}$$

We applied a statistical pooling operator $\phi_t$ that preserves both mean and covariance structure:

$$\mu^{(t)} = \frac{1}{L_t} \sum_{i=1}^{L_t} h_i^{(t)}, \quad \sum^{(t)} = \frac{1}{L_t} \sum_{i=1}^{L_t} (h_i^{(t)} - \mu^{(t)})(h_i^{(t)} - \mu^{(t)})^\top \tag{29}$$

A low-rank factorization (Nyström approximation) compresses covariance into a vector:

$$c^{(t)} = \mathrm{vec}(U_k^\top \sum^{(t)} U_k), \quad U_k \in \mathbb{R}^{d_t \times k}. \tag{30}$$

Thus, the final text embedding is

$$z^{(t)} = W_t \begin{bmatrix} \mu^{(t)} \\ c^{(t)} \end{bmatrix}, \quad z^{(t)} \in \mathbb{R}^d \tag{31}$$

## C.2 Visual Encoding

We partition an image into $L_v$ patches:

$$x^{(v)} = \{p_1, p_2, \ldots, p_{L_v}\}, \quad p_j \in \mathbb{R}^{h \times w \times c}. \tag{32}$$

A pretrained CLIP-ViT encoder $f_v : \mathcal{X}_v \to \mathbb{R}^{L_v \times d_v}$ yields patch embeddings using:

$$h^{(v)} = f_v(x^{(v)}), \quad h^{(v)} = [h_1^{(v)}, h_2^{(v)}, \ldots, h_{L_v}^{(v)}], \quad h_j^{(v)} \in \mathbb{R}^{d_v}. \tag{33}$$

Similar to the text above, we define

$$\mu^{(v)} = \frac{1}{L_v} \sum_{j=1}^{L_v} h_j^{(v)}, \quad \sum^{(v)} = \frac{1}{L_v} \sum_{j=1}^{L_v} (h_j^{(v)} - \mu^{(v)})(h_j^{(v)} - \mu^{(v)})^\top, \tag{34}$$

and compress via low-rank covariance embedding using

$$c^{(v)} = \text{vec}(U_k^\top \sum{}^{(v)} U_k). \tag{35}$$

The visual representation is then:

$$z^{(v)} = W_v \begin{bmatrix} \mu^{(v)} \\ c^{(v)} \end{bmatrix}, \quad z^{(v)} \in \mathbb{R}^d. \tag{36}$$

## C.3 Shared Latent Alignment

Both the modalities are projected into the shared latent space $\mathcal{Z}$:

$$z^{(t)} = P_t(h^{(t)}), \quad z^{(v)} = P_v(h^{(v)}), \quad z^{(t)}, z^{(v)} \in \mathcal{Z}. \tag{37}$$

We enforce distributional proximity between $(z^{(t)}, z^{(v)})$ using a contrastive alignment term:

$$\mathcal{L}_{align} = -\log \frac{\exp(\text{sim}(z^{(t)}, z^{(v)})/\tau)}{\sum_{(z^{(t)}, z^{(v')})} \exp(\text{sim}(z^{(t)}, z^{(v)})/\tau)}, \tag{38}$$

where $\text{sim}(\cdot, \cdot)$ is cosine similarity and $\tau$ a temparature parameter.

# D  Cross-Modal Attention Fusion

From the above, we obtain projected embeddings as $z^{(t)} \in \mathbb{R}^{L_t \times d}$, $z^{(v)} \in \mathbb{R}^{L_v \times d}$, where $d = 512$. We construct modality-specific query, key, and value matrices, as:

$$Q^{(v)} = z^{(v)} W_Q^{(v)}, K^{(t)} = z^{(t)} W_K^{(t)}, V^{(t)} = z^{(t)} W_V^{(t)}, \tag{39}$$

with $W_Q^{(v)}, W_K^{(t)}, W_V^{(t)} \in \mathbb{R}^{d \times d}$. Next, we defined cross-modal attention from vision to text as:

$$\alpha = \text{softmax}\left(\frac{Q^{(v)}(K^{(t)})^\top}{\sqrt{d}}\right) \in \mathbb{R}^{L_v \times L_t}. \tag{40}$$

Here, each visual token attends to all textual tokens, producing fused representations as $z^{(f)} = \alpha V^{(t)} \in \mathbb{R}^{L_v \times d}$. Unlike symmetric co-attention, this asymmetric scheme ensures that visual

grounding is enriched by linguistic semantics while avoiding representation dilution from treating both modalities equivalently. For robustness, we extended to a multi-head formulation using

$$z^{(f)} = \bigoplus_{h=1}^{H} z_h^{(f)}, \quad z_h^{(f)} = \alpha_h V_h^{(t)}, \tag{41}$$

Thus, the fused representation is a concatenation of head-specific semantic refinements. To prevent dominance of either modality, we introduced a modality gating mechanism. The scalar gate here is defined as:

$$g = \sigma(w^\top [\text{mean}(z^{(v)}), \text{mean}(z^{(t)})], \tag{42}$$

where $g \in (0, 1)$. The final fusion is a convex combination:

$$z^{(cm)} = g \cdot \text{mean}(z^{(f)}) + (1 - g) \cdot \text{mean}(z^{(t)}). \tag{43}$$

This adaptive gate balances contributions from visual-grounded fusion and raw textual semantics, ensuring stable cross-modal alignment.

## E  NEURAL MODULE ZOO AND DYNAMIC ROUTING

**Formal definition.**  Let the zoo at time $t$ contain $M_t$ active modules:

$$\mathcal{M}_t = \{m_1(\cdot; \theta_1), m_2(\cdot; \theta_2), \ldots, m_{M_t}(\cdot; \theta_{M_t})\}. \tag{44}$$

Each module is a parametric function $m_j : \mathbb{R}^d \to \mathbb{R}^d, m_j(z^{(f)}; \theta_j) = u_j$, where $u_j \in \mathbb{R}^d$ is the output embedding from module $j$. Thus, given $z^{(f)}$, the zoo produces a candidate set of transformed representations: $U = [u_1, u_2, \ldots, u_{M_t}]^\top \in \mathbb{R}^{M_t \times d}$.

**Module Families.**  The zoo supports multiple **operator families**, each corresponding to distinct inductive biases:

- **MLP modules** (dense projections):

$$m_{MLP}(z^{(f)}; \theta) = \sigma(W_2 \phi(W_1 z^{(f)} + b_1) + b_2), \tag{45}$$

  where $\phi(\cdot)$ is ReLU or GeLU, and $\sigma(\cdot)$ is a nonlinearity or identity.
- **Transformer modules** (contextual reasoning):

$$m_{Trans}(z^{(f)}; \theta) = \text{MHA}(z^{(f)}) + \text{FFN}(z^{(f)}), \tag{46}$$

  where MHA denotes the multi-head attention over $z^{(f)}$.
- **LSTM modules** (sequential bias):

$$h_t, c_t = \text{LSTM}(z^{(f)}, h_{t-1}, c_{t-1}; \theta). \tag{47}$$

- **ResNet-style modules**(residual feature refinement):

$$m_{Res}(z^{(f)}; \theta) = z^{(f)} + F(z^{(f)}; \theta), \tag{48}$$

  where $F$ is a stack of nonlinear layers.
- **Squeeze-and-Excitation modules** (channel re-weighting):

$$m_{SE}(z^{(f)}; \theta) = z^{(f)} \odot \sigma(W_2 \phi(W_1 \text{pool}(z^{(f)}))). \tag{49}$$

These families are not fixed, and new families may be introduced during evolution (see subsection 3.5). Next, to encourage **structural diversity**, each module type admits multiple instantiations with distinct hyperparameters using $\theta_j = \{W_j, b_j, \alpha_j, \dots\}$, where $\alpha_j$ represents hyperparameters such as hidden width, number of layers, or dropout rate. Let $\Omega$ denote the hyperparameter configuration space. Then for a module family $\mathcal{F}$:

$$\{m(\cdot; \theta^{(1)}), m(\cdot; \theta^{(2)}), \dots\}, \quad \theta^{(k)} \sim \Omega. \tag{50}$$

This ensures that even within the same operator family, modules exhibit functional non-redundancy, avoiding collapse into homogeneous transformations.

**Theoretical Motivation.** Given $z^{(f)}$, an optimal transformation is not known *a priori*. The zoo, therefore, acts as a **basis expansion** of nonlinear operators, where the router learns convex combinations:

$$z^{(r)} = \sum_{j=1}^{M_t} \beta_j m_j(z^{(f)}; \theta_j), \quad \beta_j \geq 0, \sum_j \beta_j = 1. \tag{51}$$

This setup can be viewed as a **functional mixture model**:

$$\mathcal{F}(z^{(f)}) \approx \sum_{j=1}^{M_t} \beta_j m_j(z^{(f)}; \theta_j). \tag{52}$$

By evolving $\mathcal{M}_t$, the model dynamically expands the representational capacity, while the router ensures sparse and efficient selection.

# F  GRAPH ATTENTION ROUTER

**Notations and Inputs.** Let the Neural Module zoo previously at time $t$ contain $N$ active modules $\mathcal{M}_t = \{m_1, m_2, \dots, m_N\}$. For a single input sample (or a batch handled elementwise), we denoted the fused embedding (router query) as $\mathbf{f} \in \mathbb{R}^d$ (from subsection 3.2), and module outputs as $\mathbf{u}_m \in \mathbb{R}^d$ for $m = 1 \dots N$. We stacked them into $U = [\mathbf{u}_1; \dots; \mathbf{u}_N] \in \mathbb{R}^{N \times d}$. Next, we implemented multi-head attention with $H$ heads; index head by $h$. Each head uses projection matrices $W_Q^{(h)}, W_K^{(h)}, W_V^{(h)} \in \mathbb{R}^{d_h \times d}$ with $d_h = d/H$.

**Headwise compatibility: relevance + synergy.** For head $h$, we computed

$$\mathbf{q}^{(h)} = W_Q^{(h)} \mathbf{f}, \quad \mathbf{k}_m^{(h)} = W_K^{(h)} \mathbf{u}_m, \quad \mathbf{v}_m^{(h)} = W_V^{(h)} \mathbf{u}_m. \tag{53}$$

We defined two components for the per-module compatibility score:

1. **query-to-module relevance** (standard scaled dot-product):

$$r_m^{(h)} = \frac{\langle \mathbf{q}^{(h)}, \mathbf{k}_m^{(h)} \rangle}{\sqrt{d_h}}. \tag{54}$$

2. **module-synergy score** that captures how module $m$ complements other modules for this input. We compute a learned module affinity via scaled dot-products on keys:

$$S_{m,j}^{(h)} = \frac{\langle \mathbf{k}_m^{(h)}, \mathbf{k}_j^{(h)} \rangle}{\sqrt{d_h}} \quad (j = 1 \dots N). \tag{55}$$

---

**Algorithm 1** GAR Forward & Bookkeeping (per batch)

---

**Require:** Fused embeddings $\{f^{(i)}\}_{=1}^B$, module outputs $U^{(i)}$, router params $\theta_r$, module params $\{\theta_m\}$

**Ensure:** Router outputs $\{z^{(r,i)}\}$, updated running stats $\{\bar{\alpha}, \Phi\}$

 1: **for** each sample $i$ **do**
 2:    **for** each head $h$ **do**
 3:       Compute $q^{(h)} = W_Q^{(h)} f^{(i)}$, $k_m^{(h)} = W_K^{(h)} u_m^{(i)}$, $v_m^{(h)} = W_V^{(h)} u_m^{(i)}$
 4:    **end for**
 5:    Compute $r_m^{(h)} = \frac{\langle q^{(h)}, k_m^{(h)} \rangle}{\sqrt{d_h}}$
 6:    Compute $S_{m,j}^{(h)}$ and $s_m^{(h)} = r_m^{(h)} + \gamma^{(h)} \cdot \sum_j \text{softmax}(S_{m,*}^{(h)}) \cdot q_{\text{int}}(S_{m,j}^{(h)})$
 7:    $\alpha_m^{(h)} = \text{softmax}_m(s_m^{(h)})$; aggregate $\alpha_m$ over heads $\rightarrow \alpha_m$
 8:    Optionally sparsify $\alpha \rightarrow \tilde{\alpha}$ (sparsemax or top-K)
 9:    $z^{(r,i)} = \sum_m \tilde{\alpha}_m \cdot v_m^{\text{agg}}$
10: **end for**
11: Compute task loss $\mathcal{L}_{\text{task}}$ using $\{z^{(r,i)}\}$
12: Compute router regularizers $\mathcal{L}_{\text{ent}}$, $\mathcal{L}_{\text{load}}$, $\mathcal{L}_{\text{budget}}$
13: Backprop: update $\theta_r$ and $\{\theta_m\}$ (with per-module LR scaling)
14: **Bookkeeping:**
15: **for** each $m$ **do**
16:    $\tilde{U}_{m,t} = \text{mean}_i \left[ \alpha_m^{(i)} \cdot (\text{baseline\_loss}_i - \text{loss}_i) \right]$
17: **end for**
18: $\Phi_m \leftarrow (1 - \eta)\Phi_m + \eta\tilde{U}_{m,t}$
19: $\bar{\alpha}_m \leftarrow (1 - \rho)\bar{\alpha}_m + \rho\,\text{mean}_i[\alpha_m^{(i)}]$
20: Send $\{\Phi_m, \bar{\alpha}_m\}$ to Evolutionary Strategist

---

The synergy was aggregated for $m$ as a normalized attention over other modules:

$$s_m^{(h)} = \sum_{j=1}^N \omega_{m,j}^{(h)} \cdot q_{int}(S_{m,j}^{(h)}), \quad \omega_{m,j}^{(h)} = \frac{\exp(S_{m,j}^{(h)})}{\sum_{k=1}^N \exp(S_{m,j}^{(h)})}. \tag{56}$$

Here, $q_{int}(\cdot)$ is an optional nonlinearity (e.g., ReLU or identity) that lets the synergy term be asymmetric and saturating if desired. We combined relevance and synergy linearly (learnable balance):

$$s_m^{(h)}(\mathbf{f}, U) = r_m^{(h)} + \gamma^{(h)} s_m^{(h)}, \tag{57}$$

where $\gamma^{(h)} \in \mathbb{R}_{\geq 0}$ is a learned (or scheduled) head-wise scalar controlling the emphasis on inter-module synergy.

**Novelty.**    The synergy term lets the router prefer modules that not only individually match the query but that form *complementary coalitions* for the current input - capturing pairwise (and via repeated application, higher-order) interactions among experts. This is distinct from class MoE routers that treat modules as independent.

**Multi-head attention and normalized routing weights.**    For head $h$, we normalized capabilities with softmax over modules:

$$\alpha_m^{(h)} = \frac{\exp(s_m^{(h)})}{\sum_{j=1}^N \exp(s_j^{(h)})}. \tag{58}$$

We aggregated heads into a single routing weight per module (head-averaging or learned projection):

$$\alpha_m = \frac{1}{H} \sum_{h=1}^{H} \alpha_m^{(h)} \quad \text{or} \quad \alpha = \text{softmax}(W_{agg}[\alpha^{(1)}; \ldots; \alpha^{(H)}]), \tag{59}$$

where $W_{agg}$ projects head-wise vectors to a final distribution is desired. Here, the output of the router is the weighted mixture:

$$\mathbf{z}^{(r)} = \sum_{m=1}^{N} \alpha_m \cdot \mathbf{v}_m^{agg}, \quad \mathbf{v}_m^{agg} = \frac{1}{H} \sum_{h=1}^{H} \mathbf{v}_m^{(h)}. \tag{60}$$

This $\mathbf{z}^{(r)}$ flows to the classification head and participates in standard backpropagation: gradients pass to $W_V, W_K, W_Q$ and - via $\mathbf{v}_m$ and $\mathbf{u}_m$ - to module parameters.

**Controlled sparsity: top-k routing (efficient, capacity-aware).** To enforce the **Max Active Modules** constraint and reduce compute, we designed **Soft $\rightarrow$ Sparse** path, where we computed dense $\alpha_m$ as above, then apply a differentiable sparsification to keep at most $K$ modules per sample. Here, we had two practical, differentiable options:

1. **Sparsemax/Entmax:** We replaced softmax with sparsemax/entmax, which produces exact zeros for many entries while remaining subgradient-based and differentiable.

2. **Gumbel-TopK with straight-through (ST) estimator:** We sampled a binary mask $g_m$ indicating top-$K$ modules (determinisitc top-$K$ at inference). During the forward pass, we used hard top-K selection:

$$g_m = \mathbf{1}\{\alpha_m \text{in top-K}\}, \quad \tilde{\alpha}_m = \frac{g_m \cdot \alpha_m}{\sum_j g_j \cdot \alpha_j} \tag{61}$$

For backprop, we used straight-through, where we propagated gradients to $\alpha_m$ as if soft selection had been used (or we kept the option of Gumbel-softmax relaxation for a differentiable approximation).

We used (and recommend) sparsemax in training for stable gradients and deterministic top-K at evaluation for reproducibility.

**Router regularizers and losses.** To prevent collapse onto a small subset of modules and to encourage exploration and load balancing, we incorporated three auxiliary terms in router training:

1. **Entropy Regularizer (exploration early in training):**

$$\mathcal{L}_{ent} = -\frac{1}{N} \sum_{m=1}^{N} \alpha_m \log(\alpha_m). \tag{62}$$

2. **Load-balancing penalty:** We encouraged average router usage $\tilde{\alpha}_m$ (running mean across samples/batches) to match uniform expectation $1/N$:

$$\mathcal{L}_{load} = \sum_{m=1}^{N} \left(\bar{\alpha}_m - \frac{1}{N}\right)^2, \quad \bar{\alpha}_m \leftarrow (1-\rho)\bar{\alpha}_m + \rho \mathbb{E}_{batch}[\alpha_m]. \tag{63}$$

3. **Sparsity budget:** If using sparsity, we penalized deviation from target active $K$ via:

$$\mathcal{L}_{budget} = \left(\frac{1}{N} \sum_{m=1}^{N} \mathbf{1}\{\alpha_m > 0\} - \frac{K}{N}\right)^2 \tag{64}$$

(or an L1 surrogate on $\alpha$).

## G  EVOLUTIONARY STRATEGIST — META-CONTROLLER FOR STRUCTURAL SELF-GROWTH

The **Evolutionary Strategist** is a meta-learning controller that continually modifies the **Neural Module Zoo** $\mathcal{M}$ during training. It operates at the level of **module genotypes** (architecture + hyperparameters) and **phenotypes** (weights, performance types), and its goal is to maximize long-term validation performance while respecting computation/complexity constraints and encouraging parametric plurality. The strategist combines: (i) an interpretable fitness signal derived from the Graph Attention Router, (ii) a set of genetic operators (prune, mutate, hybridize), and (iii) a policy $\pi_\phi$ trained with a reinforcement/meta-gradient objective. Below, we define state, actions, fitness, evolution operators, the learning objective for the controller, and practical stabilizers.

**Notation and State Representation.**   At discrete evolution decision times $t \in \{0, T_e, 2T_e, \dots\}$, the system maintains:

- Module pool: $\mathcal{M}_t = \{m_1, \dots, m_{N_t}\}$.
- Each module $m$ has:
  - genotype (hyperparameters, topology): $\theta_m$ (e.g., depth, width, dropout, heads, activation type),
  - phenotype (weights): $w_m$,
  - usage/metadata: $\text{age}_m$, $\text{params}_m$ (parameter count), $\text{FLOPs}_m$,
  - contribution statistics: tracked variables defined below.
- Global training state $S_t$ comprises:

$$S_t = \left\{ \{(\theta_m, w_m, \text{age}_m, \text{params}_m, C_m)\}_{m \in \mathcal{M}_t}, \text{val\_metrics}_{t-\Delta:t}, \text{budget\_remaining} \right\}, \quad (65)$$

where $C_m$ is a numeric contribution/fitness proxy.

The controller $\pi_\phi(a_t | S_t)$ outputs actions $a_t$ altering $\mathcal{M}_t$ (prune, spawn/mutate, hybridize, no-op, or other maintenance actions). Actions can be multi-step (e.g., hybridize two parents into one child + spawn).

**Contribution and Fitness Estimation.**   A robust, low-variance fitness signal is central. We combine two complementary, efficiently computable signals in each evolution epoch:

1. **Attention-contribution proxy (router-based)**

   For module $m$, collect the per-batch average routing weight from the Graph Attention router over a recent buffer $\mathcal{B}$ (the last $B$ mini-batches):

$$\bar{\beta}_m = \frac{1}{B} \sum_{b \in \mathcal{B}} \beta_m^{(b)}. \quad (66)$$

2. **Leave-one-out loss impact (performance-proxy)**

   For a mini-batch $b$ compute the batch loss with full routing $\mathcal{L}_{full}^{(b)}$ and the loss with module $m$ ablated (zeroing or masking its output) $\mathcal{L}_{-m}^{(b)}$. We defined per-batch delta:

$$\Delta\ell_m^{(b)} = \mathcal{L}_{-m}^{(b)} - \mathcal{L}_{full}^{(b)}. \quad (67)$$

   Positive $\Delta\ell_m$ indicates the module is helpful. The average over $\mathcal{B}$:

$$\overline{\Delta\ell}_m = \frac{1}{B} \sum_{b \in \mathcal{B}} \Delta\ell_m^{(b)}. \quad (68)$$

We combine these into an exponential moving average contribution score $C_m(t)$:

$$C_m(t) \leftarrow (1 - \rho)C_m(t - 1) + \rho \left( w_\beta \frac{\bar{\beta}_m}{\max_k \bar{\beta}_k} + w_\ell \frac{\max(0, \bar{\Delta}\ell_m)}{\max_k \max(0, \bar{\Delta}\ell_m)} \right), \qquad (69)$$

with $\rho \in (0, 1)$ smoothing factor and weights $w_\beta, w_\ell$ (0.5 each). Normalization avoids scale issues. $C_m$ is the primary short-term fitness proxy used by selection and pruning. To encourage novelty and penalize redundancy, we also compute a novelty score:

$$\text{novelty}_m = \frac{1}{N_t - 1} \sum_{k \neq m} \exp(-\gamma_\theta ||\theta_m - \theta_k||_2^2), \qquad (70)$$

and defined a combined fitness:

$$F_m = \alpha_C C_m - \alpha_{cost} \cdot \text{cost}_m + \alpha_{nov}(1 - \text{novelty}_m), \qquad (71)$$

where $\text{cost}_m$ is the normalized computational cost (params or FLOPs), and $\alpha$ are tuning scalars. Lower $\text{novelty}_m$ (i.e., more dissimilar) increases fitness via $1 - \text{novelty}$.

**Selection and Pruning**  We removed modules whose long-run contribution is consistently low while respecting stability constraints:

- **Minimum survival age:** a module must survive at least $A_{min}$ evolution intervals before being eligible for pruning.
- **Prune condition** (quantile-based):

$$\text{Prune} \quad m \quad if \quad F_m \leq Q_q(\{F_k\}_{k \in \mathcal{M}_t}) \quad and \quad age_m \geq A_{min}, \qquad (72)$$

where $Q_q(\cdot)$ is the q-th percentile ($q = 0.15$). This avoids threshold tuning across varying pool sizes. Alternatively, a dynamic threshold $\tau_t = \mu_F - \mathcal{K}\sigma_F$ can be used (recommendation).

When pruning, we first attempt **weight recycling**: if another module has an identical genotype or an identical interface, its weights may be reused or used to initialize new offspring.

**Growth (mutation) operator.**  To spawn variants, we sample parent modules according to a soft-max over fitness:

$$p_{select}(m) = \frac{\exp(\eta F_m)}{\sum_k \exp(\eta F_k)}. \qquad (73)$$

Given parent $m_p$ with genotype $\theta_p$ and weights $w_p$, we create child genotype $\theta_c$ via parameter-space mutation:

- For continuous hyperparameters (dropout, width multipliers):

$$\theta_c^{(i)} = \theta_p^{(i)} \cdot \exp(\sigma_\theta \cdot \epsilon^{(i)}), \quad \epsilon^{(i)} \sim \mathcal{N}(0, 1). \qquad (74)$$

- For discrete hyperparameters (number of heads), we applied categorical perturbation (random $\pm$ step with small probability).

Child weights are initialized by soft inheritance:

$$w_c = \gamma_{inh} w_p + (1 - \gamma_{inh})\mathcal{N}(0, \sigma_w^2). \qquad (75)$$

where $\gamma_{inh} \in [0, 1]$ controls how much of parent knowledge is retained. This reduces cold-start training and stabilizes learning when the child shares structural motifs with the parent. A growth rate constraint keeps the pool budgeted: at most $G_{max}$ new modules per evolution step and $N_t \leq N_{max}$.

**Hybridization (Co-Evolutionary Crossover)** Hybridization recombines structural motifs and hyperparameters from two high-fitness parents $m_i$ and $m_j$ to create a child $m_c$. We treat module genotypes as graph-structured objects (topology + attributes). Let $T_m = (V_m, E_m, \Theta_m)$ denote parent $m$'s topology graph, node attributes $\Theta_m$ (layer types, widths, activation), and $W_m$ the associated weight tensors.

**Crossover operator (motif splice):**

1. **Motif extraction:** We sampled subgraph $S_i \subseteq T_{m_i}$ and $S_j \subseteq T_{m_j}$ by selecting contiguous substructures using a size distribution (small-to-medium). We represent these as adjacency and attribute sets.

2. **Interface alignment:** We find interface nodes $u \in S_i, v \in S_j$ where input/output dimensionalities can be projected. If dims differ, create small projection layers $P_{in} : \mathbb{R}^{d_1} \to \mathbb{R}^{d_c}$ and $P_{out}$ as learned linear maps. This enforces compatibility.

3. **Splice:** We create child topology

$$T_c = (T_{m_i} \, S_i) \cup S_j, \tag{76}$$

where $S_j$ is grafted into $T_{m_i}$ at matched interfaces. (Symmetric alternatives allowed.)

4. **Hyperparameter recombination:** For scalar attributes in $\Theta$, we performed convex interpolation:

$$\theta_c^{(k)} = \lambda \theta_{m_i}^{(k)} + (1 - \lambda)\theta_{m_j}^{(k)}, \quad \lambda \sim \mathcal{U}(0, 1). \tag{77}$$

For categorical attributes, we used parent-sampling with probability proportional to normalized parent fitness.

5. **Weight inheritance mapping:** The parameters for retained subgraphs are copied; for grafted subgraphs, we used soft weight blending where possible:

$$W_c[shared] = \mathcal{K} W_{m_i}[shared] + (1 - \mathcal{K})W_{m_j}[shared] + \epsilon, \tag{78}$$

and new parameters are initialized as small-noise or adapted from th nearest parent via projection.

This motif-based crossover allows the child to inherit functional building blocks (e.g., a multi-head attention motif with a particular head-to-dimension ratio) and yields architectures not present in the initial search space.

**Controller Optimization** The controller $\pi_\phi$ must learn when to prune, spawn, and hybridize to maximize long-term validation performance under computation budget $B$. We pose this as a constrained expected reward maximization:

$$\max_\phi \mathbb{E}_{\tau \sim \pi_\phi} \left[ \sum_{t=0}^{T} \gamma^t r(S_t, a_t) \right] \quad s.t. \quad \mathbb{E}_{\tau \sim \pi_\phi} \left[ \text{Cost}(\tau) \right] \leq B, \tag{79}$$

where $\tau$ is an evolution trajectory, $\gamma$ discount factor, and reward $r$ is computed at evolution intervals. We used a Lagrangian relaxation:

$$\mathcal{J}(\phi, \lambda) = \mathbb{E} \left[ \sum_t \gamma^t r_t - \lambda(\text{Cost}_t - B_t) \right], \tag{80}$$

and optimize $\phi$ via policy gradient (e.g., PPO) with gradient estimator:

$$\nabla_\phi \mathcal{J} \approx \mathbb{E} \left[ \sum_t \nabla_\phi \log \pi_\phi(a_t | S_t) \tilde{A}_t \right], \tag{81}$$

---

**Algorithm 2** Evolutionary Strategist for Neural Module Evolution

---

**Require:** Module set $\mathcal{M} = \{M_1, \ldots, M_K\}$, fused embeddings $\mathbf{z} \in \mathbb{R}^d$, contribution scores $\alpha_i$, replay memory $\mathcal{R}$
**Ensure:** Updated module set $\mathcal{M}'$
1: Initialize policy $\pi_\theta$ for meta-controller
2: **while** training not converged **do**
3:     Sample task batch $\mathcal{B} \sim \mathcal{D}$
4:     Compute fused embedding $\mathbf{z}$
5:     Route $\mathbf{z}$ to modules using GraphAttentionRouter
6:     Compute contributions $\alpha_i = \mathrm{softmax}\left(\frac{\mathbf{z}^\top \mathbf{k}_i}{\sqrt{d}}\right)$
7:     Evaluate task loss $\mathcal{L}_{task}$ and reward $R(\mathcal{M}) = -\mathcal{L}_{task} + \lambda H(\alpha)$
8:     Store $(\mathbf{z}, \mathcal{M}, R)$ in replay memory $\mathcal{R}$
    {— Evolutionary Update —}
9:     **if** $\alpha_i < \tau_{prune}$ for consecutive $T$ steps **then**
10:       Remove module $M_i$ from $\mathcal{M}$                                         (Pruning Rule)
11:     **end if**
12:     **if** $R(\mathcal{M}) < \tau_{grow}$ **then**
13:       Spawn new module $M_j'$ with parameters
14:         $\Theta_j' = \Theta_j + \epsilon, \quad \epsilon \sim \mathcal{N}(0, \sigma^2 I)$                     (Growth Rule)
15:       Add $M_j'$ to $\mathcal{M}$
16:     **end if**
17:     **if** $\exists M_p, M_q \in \mathcal{M}$ with high complementarity **then**
18:       Generate child $M_c$ via crossover:
19:         $\Theta_c = \eta\Theta_p + (1 - \eta)\Theta_q, \quad \eta \sim \mathcal{U}(0, 1)$            (Hybridization Rule)
20:       Add $M_c$ to $\mathcal{M}$
21:     **end if**
    {— Meta-Controller Update —}
22:     Compute policy gradient:
23:       $\nabla_\theta J(\theta) = \mathbb{E}_{\pi_\theta}\big[\nabla_\theta \log \pi_\theta(a|\mathcal{M}) \, R(\mathcal{M})\big]$
24:     Update $\theta \leftarrow \theta + \beta\nabla_\theta J(\theta)$
25: **end while**
26:
27: **return** $\mathcal{M}'$

---

where $\tilde{A}_t$ is an advantage estimate (computed from actual validation metric improvement over a horizon $H$). The reward $r_t$ is defined as:

$$r_t = \Delta\mathrm{ValMetric}_{t \to t+H} - \eta_{comp}\Delta\mathrm{Cost}_{t \to t+H} + \eta_{div}\overline{\mathrm{overline}}_{t \to t+H}, \tag{82}$$

balancing short-term performance gain, computational cost, and architectural novelty. In practice, we set $H$ to a modest number of training steps to trade off noise vs signal. Alternatively, a meta-gradient approach can be used where action parameters are differentiable (soft choices) and the outer validation loss is differentiated w.r.t. $\phi$ by unrolling a few inner optimization steps. We recommend policy-gradient (PPO) in experiments for stability and scalability, with meta-gradient used in ablations to evaluate potential improvements.

**Stabilization, replay, and reproducibility.** Structural modifications can destabilize training. We used three stabilizers:

1. **Replay memory $\mathcal{R}$:** We maintained a buffer of representative examples (stratified by class/modality) and replay them for $R$ mini-batches immediately after structural changes. This limits catastrophic forgetting and calibrates newly created modules.

2. **Warm-start fine-tuning:** After spawning/hybridization, child modules are trained with a reduced learning rate $\eta_{child} = \zeta\eta$ for $E_{warm}$ steps before making further evolutionary decisions.

3. **Minimum-age and hysteresis:** Modules must remain for $A_{min}$ epochs to allow their contributions to be reliably estimated; pruning decisions incorporate running variance to prevent thrashing.

For reproducibility, every structural operation (prune/mutate/hybridize) is logged with a 64-bit RNG seed, parent IDs, and a deterministic construction routine. This results in reproducible architecture evolution given the same global initial seed.

## H   TRAINING OBJECTIVE

**Notation & Problem Statement.**

- Let an architecture (set of active modules and their hyperparameters) be $A = \{(m, \eta_m)\}_{m \in \mathcal{M}}$, where $\eta_m$ are module hyperparameters (depth, heads, dropout, widths), and $\mathcal{M}$ is the active module index set.

- Let $\Theta = \{\theta_m\}_{m \in \mathcal{M}}$ denote all module weights plus router and head weights; let $\theta_{ext}$ denote the multimodal extractor weights (DistilBERT, CLIP-ViT).

- Router produces per-sample soft contributions $\beta_m(x)$ for sample $x$. For a minibatch $B$, denote $\beta_m(B) = \frac{1}{|B|} \sum_{x \in \mathcal{B}} \beta_m(x)$.

- Meta-controller (Evolutionary Strategist) is parameterized by $\phi$ and implements a policy $\pi_\phi$ which, at discrete architectural decision times, outputs actions $a \in \mathcal{A}$ (prune, grow, hybridize, and their parameters).

- Let $\mathcal{R}$ be the replay buffer (capacity $N_R$).

We cast the training as the following bilevel objective:

$$
\begin{aligned}
&\text{Outer / meta (architectural) objective:} \quad \max_\phi \mathbb{E}_{\tau \sim \pi_\phi} \left[ \mathcal{P}_{val}(\Theta^\tau, A^\tau) - c \cdot \mathcal{C}(A^\tau) \right] \\
&\text{Inner / param (weights) objective:} \quad \Theta^\tau \approx \arg\min_\Theta \mathcal{L}_{train}(\Theta, A^\tau; \mathcal{D}_{train}),
\end{aligned}
\tag{83}
$$

where $\mathcal{P}_{val}$ is a validation performance metric (e.g. AUC), $\mathcal{C}(A)$ is an architectural cost (parameters, FLOPs), and $\tau$ denotes a stochastic architecture trajectory induced by $\pi_\phi$. Because architectures are discrete and evolution is online, we used a hybrid of gradient-based inner training and policy-gradient outer optimization.

**Inner (parameter) loss.**   For a minibatch $B = \{(x, y)\}$. the *base task loss* is binary cross-entropy:

$$
\begin{aligned}
\mathcal{L}_{task}(B; \Theta, A) &= \frac{1}{|B|} \sum_{(x,y) \in B} \text{CE}(y, \hat{y}(x; \Theta, A)) \\
\hat{y}(x; \Theta, A) &= \sigma(W_o z^{(r)}(x; \Theta, A)),
\end{aligned}
\tag{84}
$$

where $z^{(r)}$ is the router's weighted mixture output. To encourage *per-sample routing diversity* (avoid collapse to a single module), we used an entropy reward on router weights averaged over the batch:

$$
\mathcal{L}_{div}(B; \Theta, A) = -\frac{1}{|B|} \sum_{x \in B} \sum_{m \in \mathcal{M}} \beta_m(x) \log \beta_m(x).
\tag{85}
$$

To encourage *representational orthogonality* between module outputs (parametric plurality beyond mere usage), we included a pairwise cosine-similarity penalty:

$$
\mathcal{L}_{orth}(B; \Theta, A) = \frac{2}{|\mathcal{M}|(|\mathcal{M}| - 1)} \sum_{i < j} \left( \frac{\langle u_i(B), u_j(B) \rangle}{||u_i(B)|| \, ||u_j(B)||} \right)^2,
\tag{86}
$$

where $u_m(B) = \frac{1}{|B|} \sum_{x \in B} u_m(x)$ is the batch-averaged module output (or one can use per-sample pairwise terms averaged). We penalized *architectural complexity* (to avoid unconstrained growth):

$$\mathcal{L}_{comp}(A) = \alpha_{param} \sum_{m \in \mathcal{M}} \text{params}(m) \cdot g_m, \quad g_m = \min\{1, \text{clip}(\beta_m^{avg}/\epsilon, 0, 1)\}, \qquad (87)$$

where $\beta_m^{avg}$ is a long-run usage estimate and $g_m$ behaves as a soft gate: rarely used modules incur less cost. To mitigate catastrophic forgetting when the architecture changes, we used *replay loss:*

$$\mathcal{L}_{replay}(\mathcal{R}; \Theta, A) = \frac{1}{|\mathcal{S}|} \sum_{(x,y) \in \mathcal{S} \subset \mathcal{R}} \text{CE}(y, \hat{y}(x; \Theta, A)), \qquad (88)$$

with $\mathcal{S}$ a randomly sampled minibatch from the buffer. Finally, the inner total loss used to update $\Theta$ is:

$$\boxed{\mathcal{L}_{train}(B; \Theta, A) = \mathcal{L}_{task} + \lambda_{div}\mathcal{L}_{div} + \lambda_{orth}\mathcal{L}_{orth} + \lambda_{replay}\mathcal{L}_{replay} + \lambda_{comp}\mathcal{L}_{comp}} \qquad (89)$$

All $\lambda$'s are hyperparameters tuned to balance accuracy, diversity, and compactness. $\Theta$ is updated by standard SGD/Adam steps minimizing $\mathcal{L}_{train}$. The router parameters (and extractor finetuning) are included in $\Theta$ and receive gradients through $\beta$ and the mixture $z^{(r)}$.

**Module Fitness and Contribution Estimator.** The strategist must decide which modules to prune, which to hybridize, and which to use as parents for growth. Decisions rely on a fitness score $f_m$ per module that reflects usefulness and marginal contribution. We propose a practical estimator that balances fidelity and computation:

1. **Usage estimate (fast):**

$$u_m^{(t)} = \text{EMA}_\rho(\beta_m(B_t)), \qquad (90)$$

    an exponential moving average over minibatches with decay $\rho$.

2. **Marginal contribution (periodic, higher fidelity):** For every $T_{eval}$ minibatches, we estimated the marginal loss drop of module $m$ on a small validation probe $P$:

$$\Delta\mathcal{L}_m \approx \frac{1}{|P|} \sum_{x \in P} (\mathcal{L}(x; \Theta, A/\{m\}) - \mathcal{L}(x; \Theta, A)), \qquad (91)$$

    where $A/\{m\}$ is the architecture with $m$ ablated (set $\beta_m = 0$ and renormalize). Positive $\Delta\mathcal{L}_m$ means the module helps.

3. **Composite fitness:** We combine both signals:

$$f_m = \gamma_1 u_m^{(t)} + \gamma_2 \text{ReLU}(\Delta\mathcal{L}_m), \qquad (92)$$

    normalized across modules. $\gamma$ weights trade off frequency vs casual contribution.

The strategist prunes modules with $f_m < \tau_{prune}$ and age ¿ $A_{mini}$; spawns children from parents sampled proportional to $f_m$; selects parents for hybridization stochastically using fitness-proportionate selection.

**Evolutionary Actions.** Let action set $\mathcal{A}$ include:

- **prune(m)**: remove module $m$ permanently (or mark inactive).
- **grow**$(p, \delta_\eta)$: spawn new module from parent $p$ with hyperparameter perturbation $\delta_n$.
- **hybridize**$(p_i, p_j, \lambda)$: create child hyperparameters

$$\eta_c = \lambda\eta_{p_i} + (1 - \lambda)\eta_{p_j} + \epsilon, \epsilon \sim \mathcal{N}(0, \sigma^2). \qquad (93)$$

**Weight inheritance.** Child weights $\theta_c$ are warm-started by structured inheritance:

- for hybridization: $\theta_c = \lambda\theta_{p_i} + (1 - \lambda)\theta_{p_j} + \zeta$, with small noise $\zeta \sim \mathcal{N}(0, \sigma_w^2)$.
- for growth by mutation: copy and perturb parent: $\theta_c = \theta_p + \zeta$.

  After creation, children undergo a short warm-up period of $T_{warm}$ minibatches with a smaller learning rate $\eta_w$ to prevent destabilization.

**Knowledge distillation on pruning.** Before the pruning module $m$, we optionally perform a distillation step so that the remaining modules can absorb its functionality:

$$\mathcal{L}_{kd} = \frac{1}{|S|} \sum_{x \in S} ||z_{full}^{(r)}(x) - z_{ablated}^{(r)}(x)||_2^2, \tag{94}$$

where $z_{full}^{(r)}$ uses $m$ and $z_{ablated}^{(r)}$ does not. Minimizing $\mathcal{L}_{kd}$ for a few steps softens the removal.

**Outer (Meta) Objective and Optimization of $\phi$.** The strategist parameter $\phi$ defines a policy $\pi_\phi(a_t|s_t)$ that, given state $s_t$ (module fitness vector $\{f_m\}$, age, resource usage, recent validation trajectory, etc.), outputs an action distribution. The meta-reward $r_t$ should encourage long-term validation gains while penalizing cost:

$$r_t = \Delta\mathcal{P}_{val,t} - \eta_{param}\Delta\text{Params}_t - \eta_{flops}\Delta\text{FLOPs}_t - k \cdot \mathcal{C}_{instab,t}, \tag{95}$$

where $\mathcal{P}_{val,t} = \mathcal{P}_{val}(t + \Delta) - \mathcal{P}_{val}(t)$ is the improvement observed after applying action(s) and letting the model train for a short horizon, and $\mathcal{C}_{instab,t}$ penalizes validation volatility (to avoid reckless growth that yields unstable gains). We maximized expected return:

$$J(\phi) = \mathbb{E}_{\tau \sim \pi_\phi}\left[\sum_{t=0}^{T} r_t\right]. \tag{96}$$

We applied two practical optimization strategies here:

1. **Policy Gradient (REINFORCE).** We used sampled trajectories of length $T_{meta}$, estimate returns $R_t = \sum_{k=t}^{T} r_k$, and update:

$$\nabla_\phi J \approx \mathbb{E}\left[\sum_t \nabla_\phi \log \pi_\phi(a_t|s_t)(R_t - b_t)\right], \tag{97}$$

   where $b_t$ is a learned baseline (value network) to reduce variance. Entropy regularization $-\lambda_H \sum_t \mathcal{H}(\pi_\phi(\cdot|s_t))$ is added to encourage exploration.

2. **Truncated Meta-Gradient (Differentiable Unroll).** When computational budget allows, we unrolled $k$ inner optimization steps of $\Theta$ after an action and differentiate the validation loss w.r.t. $\phi$ via chain rule (truncated backprop through optimization). Let $\Theta_{t+k}(\phi)$ denote the inner optimized weights after $K$ steps influenced by decisions sampled from $\pi_\phi$. Then,

$$\nabla_\phi \mathcal{L}_{val}(\Theta_{t+k}(\phi)) = \frac{\partial\mathcal{L}_{val}}{\partial\Theta_{t+k}} \cdot \frac{\partial\Theta_{t+k}}{\partial\phi}, \tag{98}$$

   which we compute with automatic differentiation for small $K$. This gives lower variance but larger memory/computation. In practice, we combine both: use REINFORCE for long-horizon exploration and occasional truncated meta-gradient updates for fine-tuning.