# OpenReview forum: "Architect Thyself: Neural Darwinism and Self-Evolving Multimodal Networks"
_ICLR.cc/2026/Conference — ICLR 2026 Conference Withdrawn Submission_

### Official Review · Reviewer_ocKB · 2025-10-27

**Soundness:** 2
**Presentation:** 3
**Contribution:** 2
**Rating:** 4
**Confidence:** 4

**Summary:**

This paper proposes a meta-learned evolutionary framework, namely AI Architect Thyself, for the automatical designing of neural network structure and training hyper-parameters. Comparing with traditional AutoML methods, AI Architect Thyself treats the network structures as learnable variables that can be optimized jointly with network parameters. Specifically, AI Architect Thyself contains several key components:

(1) Multi-modal feature extractor, which use DistilBert to extract text feature and CLIP-ViT to extract vision feature.

(2) Cross-Modal Attention Fusion, which is a modified multi-head attention module to fuse the multi-modal features.

(3) Neural Module Zoo. Here the fused embeddings are used to construct a module pool, in which each module can be pruned, grown or hybridized over time, and the contribution of each module can be tracked.

(4) Graph Attention Router, which is used to guide the evolutionary network search procedure.

The experimental results on 12 multi-modal reasoning benchmarks show that the proposed method can find networks with state-of-the-art performance.

**Strengths:**

1. The proposed method treats the network structures as searchable objects, and may expand the search space comparing with traditional AutoML methods.

2. The method is well-presented.

3. The experiments cover wide-range of multi-modal tasks, and the proposed method shows good performance.

**Weaknesses:**

1. There are some important informations that are not presented in the paper. For instance, what is the initial configuration, includes the initial network structure, hyperparameters, and evolutionary seed, of the algorithm?

2. This paper claims that the proposed method dose not rely on pre-defined, human-designed search spaces. This arguement is not supported by experiments. It is better to compare the size of search spaces of the proposed method and state-of-the-art AutoML methods.

3. In the ablation study of Dynamic system configuration, there are 3 configurations, i.e. the Full Dynamic System (9 modules), No Module Pruning, and No Module Addition, result in exactly the same number of model parameters (18.7M) and Memory Usage(8.4GB), which seems unreasonable.

**Questions:**

My questions are included in the part of Weaknesses.

---

### Official Review · Reviewer_NZ8Q · 2025-10-28

**Soundness:** 3
**Presentation:** 3
**Contribution:** 3
**Rating:** 6
**Confidence:** 3

**Summary:**

Deep learning architectures are often manually designed or searched within predefined search spaces, which limits the diversity and creativity of network topologies. To unlock the potential of automated design and develop architectures that are both diverse and effective, the authors introduce a co-evolutionary hybridization engine that enables the network to design itself. Their approach jointly optimizes model parameters and a time-varying network architecture. A meta-learning controller is designed to dynamically adjust the architecture through three operations: pruning, growing, and hybridizing. Experimental results show that the proposed framework achieves superior performance across twelve multimodal datasets, consistently surpassing state-of-the-art baselines. In addition, the authors conduct ablation studies that clearly validate the effectiveness of the evolutionary operations as well as components such as cross-modal fusion and the Graph Attention Router (GAR).

**Strengths:**

1. The paper shows a degree of originality by introducing an evolutionary perspective to the dynamic design of neural networks, proposing a complete process that enables the network to self-architect, self-optimize, and self-evolve.

2. The framework is described in detail, covering every stage from input feature processing and attention design to the composition of the module zoo and the three evolutionary operators. Each component is formally defined and clearly presented, demonstrating strong methodological rigor.

3. The experiments are comprehensive and well-designed. The paper compares its approach with various prior works and consistently outperforms the state-of-the-art. It also includes systematic ablation studies to evaluate the effects of different components, such as modules from the module zoo, the attention mechanism, and the evolutionary operators. Moreover, the authors report efficiency metrics, including the number of parameters and FLOPs, which are often of practical interest to readers.

**Weaknesses:**

1. The paper lacks a figure to illustrate the proposed framework. As a result, it is difficult to intuitively understand the architecture’s evolution process, the underlying strategy, and what the final evolved model architecture looks like.

2. There is no analysis of accuracy changes during the training loop, such as how training and testing accuracy vary across different steps, epochs, or architectural states $At$ ​. Moreover, the evolution process details, such as how the contributions of different modules change across training phases, are not presented. This omission makes the framework appear as a black-box pipeline from an intuitive perspective.

**Questions:**

1. What is the fundamental difference between this work and existing NAS methods or other frameworks that dynamically grow neurons? This work seems to transform the neurons in the search space into modules, and the self-evolution process appears to perform differentiable search during training.

2. Why did the authors choose multimodal tasks involving vision and text for analysis? Would the proposed framework also be effective for other modality combinations or for unimodal tasks such as text-only or vision-only?

---

### Official Review · Reviewer_bpSB · 2025-10-31

**Soundness:** 1
**Presentation:** 2
**Contribution:** 2
**Rating:** 2
**Confidence:** 4

**Summary:**

This paper proposes a modular multimodal architecture that evolves during training via an evolutionary controller. It combines CLIP-ViT and DistilBERT features, feeds them through a learned Graph Attention Router, and periodically modifies the set of active modules through operations like crossover, mutation, and removal. The authors claim consistent gains across 12 vision-language tasks.

**Strengths:**

1. The proposed GAR router and the mechanism of tracking “module contribution” are interesting in principle.

2. Ablation studies are provided and cover different architectural components.

**Weaknesses:**

1. Evaluation setup is misleading and confusing.
The paper reports AUC, Accuracy, and F1 across all tasks—including image captioning, object detection, and retrieval datasets—without explanation. These are not the standard metrics for such tasks (e.g., CIDEr/SPICE for captioning, mAP for detection). This makes it difficult to trust the claimed improvements. It appears that incompatible tasks have been forced into a unified evaluation scheme that oversimplifies or distorts actual performance.

2. Compute claims are not properly justified.
The authors claim to train on large-scale datasets (e.g., Conceptual Captions) with a single A100 GPU in just 18 hours, while also performing evolutionary updates every 3 epochs. This seems implausible and is not supported by any runtime analysis or FLOP estimates. There’s no reporting of the cost–performance tradeoff.

3. The method largely recombines existing ideas, but the novelty is unclear.
The paper combines multiple standard components—modular sub-networks, attention-based routing, and evolutionary search—into a single framework, but does not clearly define what is novel in this combination. The descriptions of modules, routing mechanisms, and mutation strategies read more like an aggregation of known techniques than a focused innovation. Furthermore, the writing tends to overuse conceptual terms (e.g., “plurality”, “self-evolving architecture”, “cross-task generalization”) without corresponding technical specificity or empirical depth.

**Questions:**

1. Why are classification metrics used across all tasks, even when inappropriate (e.g., for captioning or detection)?
Please clarify how AUC or Accuracy were computed for MSCOCO and OpenImages, and why standard task-specific metrics were not used.

2. What is the exact search space and mutation operator definition used in the evolution strategy?
Without this, it’s difficult to know what is being learned or optimized during evolution.

3. Can you provide a cost-performance curve or FLOPs breakdown of the system over training?
Statements about efficiency and scalability cannot be validated without resource usage information.

4. What components of the proposed system are actually responsible for the gains?
Have you tested whether a simpler gating or pruning strategy could achieve similar results?

---

### Official Review · Reviewer_rLcs · 2025-11-01

**Soundness:** 2
**Presentation:** 2
**Contribution:** 2
**Rating:** 2
**Confidence:** 4

**Summary:**

The paper proposes a self-evolving neural architecture framework aiming to enable neural networks to grow, hybrid, and prune their own modules during training, claiming to overcome the limitations of traditional NAS that rely on static, human-designed architectures. The framework integrates elements from meta-learning and dynamic mixture-of-experts (MoE), and reports consistent improvements (0.9% - 4.1%) over SOTA across 12 multimodal benchmarks. However, despite the ambitious claim of building a "fully autonomous thyself neural framework," the actual implementation appears to be a module-level NAS system with limited novelty, mainly recombining pre-defined building blocks rather than autonomously creating new architectures.

**Strengths:**

1. The proposed framework is technically complete and integrates multiple existing paradigms in a coherent structure, including the notions of Parametric Plurality, Graph Attention Router, and Evolutionary Strategist. The introduction of a Module Zoo balances granularity between operator- and model-level design, supporting pruning, hybridizing, and growing operations.
2. The experiments cover 12 multimodal and vision-language benchmarks of varying sizes, demonstrating the generality of the approach. Extensive ablation studies are provided to assess the contribution of each component.

**Weaknesses:**

1. The concept of "Neural Darwinism" and "Architect Thyself," while rhetorically appealing, is not technically substantiated. The framework remains within the NAS paradigm, focusing on module-level recombination rather than the genuine self-generation of architectures.
2. The pruning/hybridizing/growing operations merely reconfigure or slightly mutate predefined modules by mixing parameters or adjusting hyperparameters. The process thus operates over a fixed template space, which contradicts the notion of "full autonomy".
3. Writing quality is suboptimal. The main framework is fragmented across sections without a clear higher-level illustration. Algorithmic pseudocode is moved to the appendix, and dependencies between components are not clearly defined. Without consulting the supplementary material, the paper is not self-contained.
4. Table 1 claims coverage of directions (LLM-based, multimodal, meta-learning, self-evolving, etc.), giving an impression of incremental integration and substantial overlap with prior work such as Dynamic MoE, MetaNAS, and EvoNAS.
5. The concept of Parametric Plurality is essentially maintaining multiple hyperparameterized variants of the same operator type; it adds diversity but lacks novelty.
6. The paper lacks theoretical grounding in terms of convergence and computational complexity analysis. No evidence is provided that the evolution process stabilizes or scales efficiently.
7. The experimental section, limited to two pages, does not analyze the evolution trajectories or compare the evolved structures with baselines. Insights into how evolution improves the model are missing.
8. The claim of "beyond human design" is not supported by any qualitative or quantitative evidence of architectures that surpass human-designed topologies.
9. No variance or statistical significance tests are reported, raising concerns about selection bias or reproducibility.

**Questions:**

- In Section 2, the dataset is formulated as $D = {(x_i^{(v)}, x_i^{(t)}, y_i)}$, but the nature of each sample component is not clearly introduced until Section 3. This late clarification affects readability and coherence. At first reading, $x^{(t)}$ could be misinterpreted as the architecture at time step $t$, due to inconsistent notation conventions.
- Regarding the Multi-Head Cross-Modal Fusion (MHCMF), the authors choose to use vision as the query and text as the key-value. While this asymmetric design can be justified, the opposite configuration (text as query, vision as key-value) could also be reasonable. No comparative experiment or ablation is provided to support this design choice.
- In Section 3.3, the definition of $m_j$ is missing or unclear when first introduced. It later becomes evident that $m_j$ denotes a module in the Neural Module Zoo, but this should have been defined explicitly upon first use.
- The mathematical notation throughout the paper is inconsistent. Some symbols are used without prior definition or are redefined later in slightly different contexts. This inconsistency significantly reduces clarity.
- The Dataset and Experimental Settings section occupies disproportionate space, while the analysis of results and the insights from the evolution process are not impressive. The experiments could benefit from qualitative visualization or evolution trajectory analysis rather than a purely tabular presentation.

---

### Note · Authors · 2025-12-04

I have read and agree with the venue's withdrawal policy on behalf of myself and my co-authors.